

# How can mountaintop $CO_2$ observations be used to constrain regional carbon fluxes?

John C. Lin[1], Derek V. Mallia[1], Dien Wu[1], Britton B. Stephens[2]

[1]Department of Atmospheric Sciences, University of Utah, Salt Lake City, Utah 84112, USA
[2]Earth Observing Laboratory, National Center for Atmospheric Research, Boulder, Colorado 80301, USA

*Correspondence to*: John C. Lin (John.Lin@utah.edu)

Manuscript Submitted to Atmospheric Chemistry and Physics

**Abstract.**

Despite the need for researchers to understand terrestrial biospheric carbon fluxes to account for carbon cycle feedbacks and predict future $CO_2$ concentrations, knowledge of these fluxes at the regional scale remains poor. This is particularly true in mountainous areas, where complex meteorology and lack of observations lead to large uncertainties in carbon fluxes. Yet mountainous regions are often where significant forest cover and biomass are found—i.e., areas that have the potential to serve as carbon sinks. As $CO_2$ observations are carried out in mountainous areas, it is imperative that they are properly interpreted to yield information about carbon fluxes. In this paper, we present $CO_2$ observations at 3 sites in the mountains of the Western U.S., along with atmospheric simulations that attempt to extract information about biospheric carbon fluxes from the $CO_2$ observations, with emphasis on the observed and simulated diurnal cycles of $CO_2$. We show that atmospheric models can systematically simulate the wrong diurnal cycle and significantly misinterpret the $CO_2$ observations, due to erroneous atmospheric flows as a result of terrain that is misrepresented in the model. This problem depends on the selected vertical level in the model and are exacerbated as the spatial resolution is degraded, and our results indicate that a fine grid spacing of ~4 km or less may be needed to simulate a realistic diurnal cycle of $CO_2$ for sites on top of the steep mountains examined here in the American Rockies. In the absence of higher resolution models, we recommend coarse-scale models to focus on assimilating afternoon $CO_2$ observations on mountaintop sites over the continent to avoid misrepresentations of nocturnal transport and influence.

## 1. Introduction

Scientific consensus among climate scientists points to carbon dioxide ($CO_2$) as the main greenhouse gas leading to climate change (IPCC, 2014). Therefore, a strong need exists to quantify and understand global carbon fluxes, among which the terrestrial biospheric component is the most dynamic, potentially even reversing signs on an annual basis from year to year (Le Quéré et al., 2015;Sarmiento et al., 2010). Yet quantifying and predicting terrestrial biospheric carbon fluxes continue to pose a challenge to researchers, as seen in the large divergence between models in projections of biospheric fluxes into the future (Cox et al., 2000;Friedlingstein et al., 2003;Arora et al., 2013) as well as in hindcast mode, particularly at the regional scale (Sarmiento et al., 2010;Stephens et al., 2007;Fisher et al., 2014).



Because hills and mountains cover almost 70% of the Earth's land surface
(Rotach et al., 2008), it is imperative to quantify and understand carbon fluxes over
"complex terrain". Case in point is the Western U.S., where significant amounts of
biomass are found above 1000 m elevation (Fig. 1). Similarly, much of the biomass and
potential for terrestrial carbon storage in other parts of the world are found in hills or
mountains, partly due to the fact that historical deforestation and biomass removal have
been most pronounced in easier-to-access, flat regions (Ramankutty and Foley, 1999).
van der Molen et al. (2007) simulated $CO_2$ variability near a Siberian
observational site and showed that even modest terrain variations of ~500 m over 200 km
could lead to considerable $CO_2$ gradients. Recently, Rotach et al. (2014) argued that
current difficulties to balance the terrestrial carbon budget are due to inabilities to handle
complex terrain. While these authors presented a strong case for the consideration of
flows over complex terrain, they did not quantify the implications of neglecting such
flows for interpreting $CO_2$ observations.
Despite the importance of regions with complex terrain in regional to global
carbon cycling, these areas have hitherto been under-sampled due to logistical difficulties,
harsh environmental settings, and violation of flat terrain assumptions in eddy covariance.
However, the significance of complex terrain has led to efforts to start closing this gap, in
regions such as Europe (Pillai et al., 2011) and the American Rockies ((Schimel et al.,
2002); see below).
The American Rockies will be the focus region of this study, which attempts to
show how $CO_2$ concentrations in mountain regions can be properly linked, through
atmospheric transport, to biospheric fluxes. While the objective of this paper is to use the
American Rockies as a case study to illustrate general aspects of interpreting $CO_2$
observations in mountainous regions, several other compelling reasons exist for studying
this region. Both models and observations have suggested that significant carbon storage
can occur in the American Rockies (Fig.1) (Schimel et al., 2002;Monson et al.,
2002;Wharton et al., 2012), albeit this storage is highly sensitive to environmental drivers
such as temperature and water availability (Monson et al., 2006;Schwalm et al.,
2012;Wharton et al., 2012;Potter et al., 2013) as well as disturbances such as insect
infestation (Negron and Popp, 2004) and wildfires (Wiedinmyer and Neff, 2007). These
disturbances are also coinciding with rapid population increases in this region (Lang et al.,
2008), with concomitant rise in urban $CO_2$ emissions (Mitchell et al., In Review), urban-
wildland interfaces (Mell et al., 2010), and demands for water resources (Reisner,
1993;Gollehon and Quinby, 2000).
Recently, several research efforts have attempted to improve the understanding of
carbon fluxes in the American Rockies. Direct eddy covariance-based measurements of
carbon fluxes have been carried out in the mountains (Blanken et al., 2009;Yi et al.,
2008); however, the eddy covariance technique characterizes fluxes only over a small
area of ~1 km$^2$ (Baldocchi et al., 2001) and requires careful attention to potential biases
from local advection. Ground-based ecological measurements (Anderegg et al.,
2012;Tkacz et al., 2008) yield detailed information regarding the ecosystem, but such
observations are also limited in spatial coverage and temporal resolution. Atmospheric
$CO_2$ observations can characterize fluxes over hundreds of km (Gerbig et al., 2009),
providing important regional scale constraints. Aircraft-based $CO_2$ measurements in this
region have had some success in characterizing regional scale fluxes (Desai et al., 2011),



albeit on a sporadic, campaign-based setting.  More significantly, a network of accurate
$CO_2$ observations has been maintained on mountaintops in the Rockies for the past
decade (Stephens et al., 2011).  These observations have been assimilated by
sophisticated global carbon data assimilation systems such as "CarbonTracker" (Peters et
al., 2007) to retrieve biospheric carbon fluxes over the mountainous regions and the rest
of the globe.
Due to the expanding number of $CO_2$ observations in mountainous areas and the
need to understand carbon fluxes in such regions, a strong motivation exists to evaluate
existing methods in which $CO_2$ observations are used in atmospheric models to retrieve
carbon fluxes.  We specifically adopt the observed diurnal cycle during the summer
growing season as a key diagnostic to evaluate models.  This is because the diurnal cycle
during the growing season, with nighttime respiratory release and daytime photosynthetic
drawdown of $CO_2$, is a prominent feature in the coupling between biospheric fluxes and
the atmosphere and one of the dominant modes in the $CO_2$ time series (Bakwin et al.,
1998;Denning et al., 1996). Furthermore, models tend to either use $CO_2$ data from the
nighttime (Keeling et al., 1976) (to sample subsiding air in the mid-troposphere) or from
the daytime (during well-mixed conditions), and aspects of the diurnal cycle can provide
clues as to whether the model is capturing the link between fluxes and concentrations
right at either, both, or neither of these times.
The diurnal pattern of $CO_2$ observed at the Storm Peak Laboratory, Colorado, was
examined by one of the first mesoscale modeling studies that investigated the impact of
mountain flows on $CO_2$ concentrations (De Wekker et al., 2009).  Although this study
adopted an idealized simulation covering only a single day of observations, it nonetheless
underscored the role of daytime upslope winds.  A common approach is to assimilate
mountain observations at night (Peters et al., 2007), favoring subsidence conditions
characterizing free tropospheric concentrations and avoiding the need to resolve daytime
upslope flows (Keeling et al., 1976).
Recently, Brooks et al. (2016) used pseudo-observations to examine the
detectability of a regional flux anomaly by three mountaintop $CO_2$ sites in the American
Rockies (including Storm Peak Laboratory).  For the atmospheric model they adopted a
time-reversed Lagrangian particle dispersion model (LPDM), which yields the
"footprint", or source region, of the observation sites (Lin et al., 2012).  Although this
study investigated whether the three mountaintop sites could detect signals of ecosystem
disturbance, Brooks et al. (2016) did not specifically examine issues related to erroneous
atmospheric transport in complex terrain nor compare modeled $CO_2$ against observed
values.
In this paper, we will focus on the same 3 mountaintop $CO_2$ sites in the American
Rockies and specifically examine the implications of using nocturnal versus daytime data
within models, in light of atmospheric models at various grid spacings—from high
resolution regional simulations to coarser global scale simulations.  More specifically, we
will drive a time-reversed LPDM with various meteorological fields and receptor heights.
We will probe the implications on the footprint, transport, and the resulting $CO_2$
concentrations as the driving meteorological fields are degraded with coarser grid spacing
and also as different vertical levels within the model are used.
The guiding questions of this paper are, as follows:





1. How do atmospheric flows in mountainous areas affect $CO_2$ concentrations and their
representation in models?
2. What are the errors incurred due to the use of coarse-scale atmospheric simulations?
3. How can mountaintop $CO_2$ observations be used in an effective manner to constrain
regional carbon fluxes in complex terrain?
**2.  Methodology**
**2.1  RACCOON Observations**
The Regional Atmospheric Continuous $CO_2$ Network (RACCOON,
http://raccoon.ucar.edu) was established in 2005 and has collected in situ $CO_2$
measurements at up to six sites over the past decade (Stephens et al., 2011). Here we
present and simulate observations from the three longest running high-alpine sites (Fig. 2;
Table 1). The easternmost site (NWR) is at 3,523 m elevation near the treeline on Niwot
Ridge, just west of Ward, CO. Niwot Ridge is a LTER site and there is an AmeriFlux
tower run by the University of Colorado 3 miles east and 500 m lower on the ridge. The
instrumentation reside in the "T-Van" where the U.S. National Oceanic and Atmospheric
Administration (NOAA)'s Global Monitoring Division has collected weekly flask
samples for measurement of $CO_2$, isotopes, and other species for over 40 years, and daily
flasks since 2006.  The middle site (SPL) is at the Desert Research Institute's Storm Peak
Lab (3,210 m on Mt. Werner near Steamboat Springs, CO). This mountaintop
observatory has a long history of measurements related to cloud physics, cloud-aerosol
chemistry, and air quality. The westernmost site (HDP) is on Hidden Peak (3,351 m,
above the Snowbird ski resort, Utah). This mountaintop site generally experiences
regionally well-mixed or free-tropospheric air, but with influences from Salt Lake City
during boundary-layer growth and venting periods.
The RACCOON measurements are based on a LiCor LI-820 single-cell IRGA
with frequent calibrations.  The instruments sample air from one of three inlet lines on a
tower (two at HDP) and use a suite of four calibration gases plus a fifth surveillance gas.
All reference gases are rigorously tied to the WMO $CO_2$ Calibration Scale with use of the
NCAR $CO_2$ and $O_2$ Calibration Facility.  100-second average measurement precision is ±
0.1 ppm (1 $\sigma$), and intercomparability is estimated from several methods to be 0.2 ppm
(Stephens et al., 2011).
We filtered out observations in which the within-hour standard deviation is
greater than 1.0 ppm or when the differences between the top two inlets are greater than
0.5 ppm, which indicate periods when significant influences that are highly localized to
the site are affecting the observations.  This filtering removed 15%, 16%, and 27% of the
hourly observations at HDP, SPL, and NWR, respectively.  Regardless, filtering made
negligible differences in the observed diurnal cycles in $CO_2$ (see Supplement).
Henceforth, we will refer to the filtered observations when discussing the observed $CO_2$.
**2.2  WRF-STILT Atmospheric Model**
The atmospheric modeling framework adopted in this study is a Lagrangian time-
reversed particle dispersion model, the Stochastic Time-Inverted Lagrangian Transport
(STILT) model (Lin et al., 2003), driven by a mesoscale gridded model, the Weather
Research and Forecasting (WRF) model (Skamarock and Klemp, 2008).  STILT is a



Lagrangian model that simulates the effects of turbulent dispersion using the stochastic
motions of air parcels. It has been widely applied to the interpretation of $CO_2$ and trace
gases in general (Lin et al., 2004;Hurst et al., 2006;Göckede et al., 2010;Kim et al.,
2013;Mallia et al., 2015;Jeong et al., 2012). WRF is a state-of-the-art non-hydrostatic
mesoscale atmospheric model that can simulate a variety of meteorological phenomena
(Skamarock and Klemp, 2008), gaining widespread acceptance and usage among the
atmospheric science community. Careful coupling between WRF and STILT has been
carried out, with an emphasis towards physical consistency and mass conservation
(Nehrkorn et al., 2010).
For this study, we ran WRF in a nested mode centered between Utah and
Colorado where the RACCOON sites are located (Fig. 2). The grid spacing was refined
in factors of 3, from 12 km grid spacing covering the entire Western U.S. to 4 km and
then to 1.3 km in the innermost domain that covers all of the RACCOON sites. 41
vertical levels were adopted, with 10 of these levels within 1 km of the ground surface,
following Mallia et al. (2015). Comprehensive testing of different WRF settings—e.g.,
convection, microphysical, and planetary boundary layer (PBL) schemes—have been
carried out as part of a previous publication (Mallia et al., 2015), and we refer the reader
to that paper for details regarding the WRF configuration. In addition to the testing
reported in Mallia et al. (2015), we have also carried out evaluation of the WRF fields
specifically using meteorological measurements on mountaintops, near the RACCOON
sites. These evaluations reveal that errors in the simulated meteorological fields are
reasonable when compared against other atmospheric simulations evaluated in less
complex terrain (Mallia et al., 2015), and biases are especially small for the WRF-1.3km
fields (Table 2). In this paper, we will examine the resulting differences in
meteorological and $CO_2$ simulations when STILT is driven by WRF fields at three
different grid spacings.
In addition to the three WRF domains, we drove STILT with a fourth
meteorological field, from NCEP's Global Data Assimilation System (GDAS). GDAS is
archived at $1^{o} \times 1^{o}$ grid spacing, at 6 hourly intervals and at 23 vertical pressure levels.
Driving STILT with GDAS was a means by which we attempted to construct an
atmospheric model to resemble the NOAA CarbonTracker product, which was also at
$1^{o} \times 1^{o}$ resolution (and 25 vertical levels) over North America. More details about
CarbonTracker can be found in the next section.
Driven by the various meteorological fields, STILT released 2000 air parcels
every 3 hours (00, 03, 06, …21 UTC) for the months of June, July, and August 2012
from the RACCOON sites and transported for 3 days backward in time. An example of
STILT-simulated air parcel trajectories can be found in Fig. S1. The choice of 2000
parcels followed from results from sensitivity tests in a previous study, also over the
Western U.S. (Mallia et al., 2015). In the case of WRF, STILT has the capability to
transport the parcels in a nested fashion. So when we refer to "WRF 1.3km simulations",
it actually means that the atmosphere in the innermost domain (Fig. 2) was simulated at
1.3 km, switching to 4km grid spacing when the parcel left the 1.3km domain; likewise,
the 12km winds were used when the parcel left the 4km domain. For the "WRF 4km
simulations" we started with the 4 km fields as the innermost domain, and then 12 km in
the outer domain.





223   For each site, we released STILT parcels using two different ways to determine
224 starting levels. When we refer to "AGL", we mean that the starting height was set at the
225 level of the inlet above the ground surface (Table 1), following the local terrain as
226 resolved in the meteorological model (whether at 1.3-, 4-, 12-km, or $1^{o}$ grid spacing).
227 The alternative method, referred to as "ASL", means that the starting level was set to the
228 elevation above sea level. For instance, the HDP site is located at 3351 m above sea level.
229 The ground height as resolved by the 12km WRF model is at 2357 m, so the starting
230 height was placed at 994 (=3351 – 2357) m above the resolved terrain. CarbonTracker,
231 as well as many other global-scale models (Geels et al., 2007;Peters et al., 2010) places
232 the observation site at an internal model level following the ASL method, so the "GDAS-
233 ASL" runs were a means by which we attempted to mimic the global model configuration
234 and to illuminate potential errors that could result from such a configuration. We also
235 tested the AGL height for GDAS, at HDP only. As shown later, these runs were highly
236 erroneous, so we did not carry them out for the other two sites.
237   The STILT-simulated air parcels were tracked as they were transported
238 backwards in time from the RACCOON receptors (see example in Fig. S1); when they
239 were in the lower part of the PBL, the locations of the parcels and amount of time the
240 parcels spend in the lower PBL were tallied. This information was used in calculation of
241 the "footprint"—i.e., the sensitivity of the receptor to upwind source regions (in units of
242 concentration per unit flux). For more details, see Lin et al. (2003). The footprints,
243 encapsulating the atmospheric transport information, were then combined with gridded
244 fluxes from the biosphere and anthropogenic emissions, which are described in the next
245 sections.

### 246 2.3 CarbonTracker $CO_2$ Concentrations and Biospheric Fluxes

247   CarbonTracker is a carbon data assimilation system covering the whole globe that
248 retrieves both oceanic and terrestrial biospheric carbon fluxes (Peters et al., 2007).
249 Observed atmospheric $CO_2$ concentrations are assimilated by CarbonTracker, which
250 adjusts carbon fluxes to minimize differences with the observed $CO_2$ using an ensemble
251 Kalman filter methodology.
252   We took three-dimensional $CO_2$ fields from CarbonTracker to initialize $CO_2$
253 concentrations at the end of the 3-day back trajectories from STILT. CarbonTracker-
254 derived biospheric fluxes, along with anthropogenic and fire emissions (Sect. 2.4), were
255 also multiplied with STILT-derived footprints and combined with the initial $CO_2$
256 concentrations to yield simulated $CO_2$ at the RACCOON receptors.
257   CarbonTracker is maintained and continues to be developed by the NOAA's Earth
258 System Research Laboratory. For this paper, we adopt the "CT-2013b" version. CT-
259 2013b provides multiple prior estimates of the oceanic, terrestrial, and fossil carbon
260 fluxes, with each combination yielding separate posterior fields of carbon fluxes and $CO_2$
261 distributions. CT-2013b results are presented as an average across the suite of prior
262 fluxes and $CO_2$ fields.
263   CT-2013b resolves atmospheric transport and fluxes at $1^{o} \times 1^{o}$ over North America
264 and $3^{o}$-lon $\times$ $2^{o}$-lat in the rest of the globe, with 25 vertical levels. The driving
265 meteorological fields come from the European Centre for Medium-Range Weather
266 Forecasts's ERA-interim reanalysis. The ensemble Kalman filter system within
267 CarbonTracker solves for scaling factors on weekly timescales to adjust upward or
268 downward biospheric carbon fluxes. The adjustments were made over "ecoregions" on





land, rather than attempting to adjust fluxes within individual gridcells, as way to reduce
the dimensions of the inversion problem within CarbonTracker. More details regarding
the CarbonTracker system can be found in Peters et al. (2007) and on-line at
http://www.esrl.noaa.gov/gmd/ccgg/carbontracker/CT2013B/.
Since CarbonTracker was designed for global carbon cycle analyses to retrieve
large-scale fluxes, the adjustment to biospheric carbon fluxes could result in artifacts at
the local to regional scales. More specifically, the attempt to match $CO_2$ observations
with a single scalar can result in flipped diurnal cycles, causing carbon uptake during the
night that is partly offset by enhanced respiration in a nearby ecoregion (Fig. S2). For this
paper, we implemented a fix that removed this artifact, preserving the 24-hour integrated
carbon flux (Fig. S3).
**2.4 Anthropogenic and Fire Emissions**
Anthropogenic $CO_2$ emissions were obtained from the Emission Database for
Global Atmospheric Research (EDGAR) (European Commission, 2009), which resolves
emissions globally at 0.1°×0.1° annually. In order to temporally downscale the annual
emissions, hourly scaling factors were obtained from the Vulcan emission inventory
(Gurney et al., 2009) and applied to the EDGAR annual emissions. Lastly, $CO_2$ emissions
from EDGAR for Year 2010 were extrapolated to 2012 using population growth rates
across the U.S. since 2010, as this was the last year in which EDGAR emissions were
available.
Wildfire emissions for $CO_2$ were obtained from the Wildland Fire Emissions
Inventory (WFEI) (Urbanski et al., 2011). Since these emissions were only reported daily,
three-hourly diurnal scaling factors were obtained from Global Fire Emissions Database
v3.1 and applied to the daily WFEI emissions to downscale the emissions to sub-daily
timescales (Mu et al., 2011;van der Werf et al., 2010).
Contributions from anthropogenic and wildfire emissions, on average, to the mean
$CO_2$ diurnal cycle observed at all the mountain sites were secondary in comparison to the
biosphere (Fig. S4). In particular, the wildfire contributions were episodic and averaged
out to negligible contributions over Jun~Aug 2012. Because of this, we will not touch
upon wildfires in the remainder of the paper.
**3. Results**
**3.1 Observed versus Simulated Diurnal Cycle**
The observed and simulated diurnal cycles of $CO_2$ for the three selected RACCOON sites
are shown in Fig. 3. The observed diurnal cycle exhibits an amplitude of ~2 ppm, on
average, with more elevated concentrations at night and depleted values during the day.
In contrast to the observed diurnal cycles, the simulated $CO_2$ extracted from
CarbonTracker output exhibits a different cycle. Instead of peaking at night, $CO_2$ in
CarbonTracker reaches its maximum during the afternoon at HDP. At SPL and NWR,
the diurnal cycle is significantly attenuated, with nighttime values barely elevated over
the background instead of the nighttime enhancement in the observed values.
It appears that the erroneous diurnal pattern at HDP within CarbonTracker can
partly be due to the diurnal reversal in the original biospheric fluxes, which showed
strong uptake of $CO_2$ even at night for the gridcell where HDP is located (Fig. S2). This



resulted in erroneous diurnal patterns at all of the lowest 8 levels of CarbonTracker (Fig.
S5), with the bottom 2 levels exhibiting strong depletions in $CO_2$ at night and
enhancements during the day, pointing to unrealistic nighttime uptake and daytime
release.
However, the diurnal reversal in biospheric fluxes alone does not completely
explain the erroneous diurnal pattern. Differences in the diurnal pattern between GDAS-
ASL simulations after introducing the diurnal fix in biospheric fluxes were not as
pronounced at SPL and NWR.
The GDAS-ASL simulations show a pronounced peak of $CO_2$ in the morning that
is missing from observations at all three sites (Fig. 3). We will discuss this feature, also
seen in other coarse-scale simulations of mountaintop $CO_2$ (Geels et al., 2007), in Sect.
3.2 below.
In contrast to GDAS-ASL and CarbonTracker, the WRF-driven simulations better
reproduce the shape of the observed diurnal cycle (Fig. 3), with nighttime enhancements
and daytime depletions of $CO_2$. Considerable differences in nocturnal $CO_2$
concentrations are found, however, in the WRF-STILT runs at various grid spacings.
WRF-12km significantly overestimates $CO_2$ at night, while WRF-1.3km and -4km
produced similar $CO_2$ concentrations that correspond much more closely to observed
values. While GDAS simulations started near the ground ("GDAS-AGL") also exhibit
nighttime enhancements and daytime depletions of $CO_2$, the nighttime values are grossly
estimated, exceeding even the values in WRF-12km. Therefore, we do not present
GDAS using the AGL configurations at the other two sites.
Part of the error in all the simulations against the observations could arise from
errors in the CarbonTracker boundary condition imposed at the end of the STILT back
trajectories. Evaluations of CT-2013b against aircraft vertical profiles (which were not
assimilated into CarbonTracker) at the Trinidad Head and Estevan Point sites on the West
Coast of the North American continent carried out by the CarbonTracker team
(http://www.esrl.noaa.gov/gmd/ccgg/carbontracker/CT2013B/profiles.php) indicate that
CT-2013b overestimates $CO_2$ concentrations by at most 1.0 ppm, on average, during the
summer season. Thus, the fact that GDAS and CarbonTracker underestimate $CO_2$ at
night likely cannot be attributed solely to a biased boundary condition.
**3.2 Differences in Simulated Transport to Mountaintop Sites**
*3.2.1 Footprint Patterns*
In order to isolate the impact of differences in atmospheric transport on the simulated
$CO_2$, we examine the average diurnal pattern of the footprint strength over Jun~Aug
2012 (Fig. 4). At each hour of the day we summed the spatially explicit map of the
average footprint that marks out the source region of each RACCOON site—shown in
Figs. 5~6 for HDP and in the Supplemental Information for the other 2 sites. The result
shows the diurnal pattern of the sensitivity of the receptor concentration to upwind fluxes.
To a large extent, the diurnal variation in footprint strength mirrors the simulated
$CO_2$ concentrations. Nocturnal enhancements in the footprints are seen in the WRF-
driven simulations, with the WRF-12km exhibiting the strongest nocturnal footprints.
Footprints from WRF-1.3km and WRF-4km are weaker at night than from WRF-12km
and closely resemble each other. GDAS-AGL footprints (only shown at HDP) are the
highest among all models at night, leading to the drastic overestimation in $CO_2$ in Fig. 3.



In contrast, GDAS-ASL footprints exhibit a peak in the morning and are generally
smaller in value than their WRF counterparts at other times of the day at HDP and NWR.
At SPL, the GDAS-ASL footprint strengths are stronger and more in line with values
from the other models.
Footprints are weaker during the daytime, and in contrast to the nighttime,
differences between footprint strengths simulated by different models are significantly
smaller. In particular, the differences are minimized in the afternoon.
These patterns are also seen in the footprint maps. We further examine
differences in the spatial patterns of average footprints produced from the various WRF
and GDAS configurations. The spatial patterns are contrasted at two different times of
the day, associated with the nighttime and afternoon hours: 0200MST (0900UTC) and
1400 MST (2100UTC), respectively. Only HDP is shown for these two hours of the day
in Figs. 5 and 6; similar figures for SPL and NWR can be found in the Supplementary
Information (Figs. S6~S9). The footprint maps show marked differences at night (Fig. 5):
the WRF-12km footprints are clearly stronger than their counterparts from the other 3
model configurations, with higher values covering the Wasatch Range near the HDP site.
Meanwhile, the GDAS-ASL footprint at 0200 MST shows a striking contrast, with very
low values around HDP and the Wasatch area in general.
The afternoon footprints at 1400MST (Fig. 6) display much more similarity with
each other. Not only do the spatial patterns between the WRF and GDAS runs resemble
one another; the significant differences in footprint strengths, with overestimation by
WRF-12km and underestimation by GDAS-ASL, are no longer found. The
aforementioned nighttime divergence and afternoon correspondence between footprint
patterns are repeated at the SPL and NWR sites (Figs S6~S9).
To further understand the nighttime divergence between model configurations, we
now examine the average air parcel trajectories within Figs 5 and 6. It is worth noting
that these trajectories differ from conventional mean wind trajectories that do not
incorporate effects from turbulent dispersion (Lin, 2012). Instead, these mean trajectories
are determined by averaging the 2000 stochastic air parcel trajectories from STILT used
for simulating transport arriving at a specific hour at a particular site, and then averaging
over the ~90 days spanning June~August 2012. Thus there are ~180,000 stochastic
trajectories averaged into generating the mean trajectory, thereby incorporating the net
effect of turbulence on atmospheric transport. An example showing a subset of stochastic
air parcels giving rise to the average trajectory is given in Fig. S1 for NWR, for 1400
MST.
Similar to the footprints, average trajectories differ much more at night than in the
afternoon. Differences in average air parcel trajectories and the underlying resolved
mountainous terrain are further examined in the next section.

### 3.2.2 Three-dimensional Terrain and Trajectories

The 3D terrain plots in Figs. 7, 9, and 11 illustrate the degradation in terrain
resolved by coarser grid spacings and the resulting differences in average STILT-derived
stochastic air parcel trajectories started at night (0900UTC) from the three sites. The
afternoon (2100UTC) plots are shown in the Supplementary Information (Figs. S10~S12).
The PBL heights, which determine whether air parcels are affected by surface fluxes (and
lead to nonzero footprint values) are also plotted as blue lines in the same plots. Note





that the apparent intersection of the PBL height with the ground in Figs. 7 and 9 is an
artifact from averaging of multiple PBL heights along stochastic trajectories (Fig. S1).
Despite terrain smoothing compared against WRF-1.3km, WRF-4km produced
STILT trajectories that are very similar to those from WRF-1.3km at all three sites,
suggesting that salient features of the mountain flows resolved with 1.3km spacing are
also found in the 4km spacing. In contrast, WRF-12km and GDAS-ASL both differed
significantly from the more finely-gridded WRF simulations.  Not only did the
trajectories deviate from the higher resolution counterparts; the relationships between the
trajectory vis-à-vis the PBL height, critical for determining footprints and simulating $CO_2$
changes (Sect. 2.2), also differ.  The WRF-12km trajectories spend more time within the
PBL, while GDAS-ASL trajectories are found much less within the PBL, because they
start at a greater height above ground level.
An alternative perspective is to view the trajectory and PBL heights relative to the
ground surface ("AGL") instead of above sea level, at each time step backward in time
from the receptor (Figs. 8, 10, 12).  These figures highlight the fact that while PBL
dynamics in the three WRF configurations are similar, the heights of the trajectories
relative to the PBL height differ.   The trajectory exits above the nocturnal PBL one hour
backward in time, on average, while the WRF-12km trajectory spends several hours
within the PBL.
The difference in the trajectory behavior can be explained by the differing terrain.
In mountainous terrain, PBL heights generally follow the terrain elevations, albeit with
attenuated amplitude (Steyn et al., 2013).  Thus in WRF-1.3km and 4km, the more highly
resolved terrain produced shallow nocturnal PBL height that descend in the valley (Fig. 7)
while the corresponding trajectory hovers above it.  Viewed relative to the ground surface
(Fig. 8), the trajectory originating from HDP appears to have exited above the nocturnal
PBL in WRF-1.3km and 4km.  In contrast, due to the significantly "flattened" mountains
in WRF-12km and in GDAS, the PBL heights exhibit less spatial variation near the
mountaintop receptor, since the terrain itself was smoothed.  Consequently,  WRF-12km
trajectories, unlike the WRF-1.3km or -4km cases, travel closer to the ground surface,
within the nighttime PBL, even as it is advected away from the three RACCOON sites
(Figs. 7, 8).  This resulted in stronger nighttime footprints in WRF-12km as seen in Figs.
4 and 5.  Another effect of the proximity of the air parcels to the model's ground surface
is the slower windspeeds from surface drag, causing the air parcel trajectories to remain
close to the 3 sites until the previous day; for HDP and SPL, the mean trajectories spiral
toward the site at the surface, following an "Ekman wind spiral" pattern (Holton, 1992).
In WRF-1.3km or WRF-4km, the measurement sites are at significantly higher elevations
above the resolved valleys in the area surrounding the sites, and the air parcels are found
above the shallow nocturnal boundary layer hugging the valley floor, on average (Fig. 7).
Although both WRF-12km and GDAS poorly resolve the mountains, a key
difference in the case of GDAS-ASL is that the air parcels were released at a site's
elevation above sea level (following what is generally done in CarbonTracker, and other
global models), much higher above ground than the release used in WRF-12km, which
was selected to be the height in AGL above the flattened mountain. Therefore, the
GDAS-ASL trajectories were significantly higher than the PBL height in the model
(particularly at HDP and NWR), which followed the flattened ground surface in the $1^{o} \times 1^{o}$
grid spacing.  Another noticeable difference in GDAS-ASL trajectory was the



significantly higher daytime PBL heights (Figs. 8, 10, 12). We suspect this is because of
the greatly reduced vertical resolution within GDAS (23 levels versus 41 levels in WRF):
since STILT diagnoses the PBL height to correspond to a model level, a higher PBL
height was chosen for GDAS because of the thicker vertical level. Another subtle artifact
of the coarse resolution within GDAS can be seen in the anomalously low daytime PBL
height just in the vicinity of HDP (Figs. 13, S10). It appears that the GDAS model set an
entire $1^{o} \times 1^{o}$ grid box near HDP to be water body (the Great Salt Lake), thereby
suppressing the PBL height.
The three-dimensional plots can explain the higher nighttime footprint strengths at
SPL (Figs. 4, S6). This result appears to be a consequence of the relative elevation of the
site and surrounding terrain. The elevation of the surrounding valley floor at SPL is
closer to that of the mountaintop location of SPL (Fig. 9); therefore, air parcels released
from SPL would have a stronger tendency to reside within the PBL even over the
surrounding valleys, unlike the steeper dropoff--i.e., deeper valley--upwind of HDP (Fig.
7) and NWR (Fig. 11).
As already found in the footprints (Fig. 5), the afternoon (2100 UTC) differences
in air parcel trajectories are much smaller (Figs. S10~S12). We suspect that this is due to
the fact that the deeper daytime PBL height causes the trajectories to reside within the
PBL, and stronger mixing within the daytime PBL minimize the relative terrain
differences. A previous modeling study focusing on the SPL area has also suggested the
daytime afternoon PBL depth to extend above the mountaintop (De Wekker et al., 2009),
indicating that differences between terrain resolution and the resulting flows could be
reduced due to the strong mixing taking place within the deep afternoon PBL.
Consequently, simulations in the afternoon show much smaller divergence between
various model configurations, resulting in similar footprint strengths and $CO_2$ values
(Figs. 3 and 4). More evidence of the convergence in afternoon simulated $CO_2$ can be
found in the small differences in $CO_2$ modeled at CarbonTracker's different levels during
this time (Fig. S5).
A few studies have specifically focused on the flows and atmospheric transport
around the NWR site. These authors have pointed to thermally driven flows, particularly
downslope drainage flow events at night (Sun et al., 2007;Sun and De Wekker,
2011;Blanken et al., 2009). Daytime upslope events, while weaker, were also noted (Sun
and De Wekker, 2011;Blanken et al., 2009;Parrish et al., 1990). It may seem that the 3D
trajectories in Fig. 11 and Fig. S12 run counter to the presence of such thermally driven
flows. We suspect that this is because the thermally driven flows induced by the terrain
cannot be discerned in the mean trajectories, which also reflect the larger scale flows that
can be stronger than the local scale thermally driven flows (Zardi and Whiteman, 2013).
When one examines the stochastic trajectories from which the mean trajectories are based
(Fig. S1), it is clear that some upslope trajectories can be detected.
We now examine the reason for the erroneous daytime peak in simulated $CO_2$
from GDAS-ASL that does not show up in the observations (Fig. 3). We specifically
focus on this feature because the daytime peak was also found in other coarse-scale
simulations of $CO_2$ for mountaintop sites--e.g., in Europe (Geels et al., 2007). Focusing
on the three-dimensional plots at the hours of 0800 and 1100 MST (Fig. 13), when the
simulated peaks are found at SPL and both NWR/HDP, respectively, the peaks coincide
with times when average trajectories are found within a relatively shallow morning PBL.





As the air parcels move backward in time, when the morning transitions backward in
time to the nighttime, many of them would still be found within the shallow nighttime
PBL.  Due to the shallowness of the nocturnal PBL, the footprint values for the air
parcels found there would be high.  These parcels would also be sampling the nighttime
$CO_2$ release and therefore lead to enhancements in $CO_2$.  In other words, the erroneous
daytime peak reflects enhanced $CO_2$ that is vented up to the observing height within the
model during the day.  We suspect that something similar is taking place in other global
models, leading to similar erroneous daytime $CO_2$ peaks (Geels et al., 2007).
**4. Discussion**
This study has sought to answer the question:  how can mountaintop $CO_2$ observations be
used to constrain regional scale carbon fluxes, given the complex terrain and flows in the
vicinity of mountaintop sites?  To address this question, we have driven a Lagrangian
particle dispersion model simulating the transport of turbulent air parcels arriving at 3
mountaintop $CO_2$ sites in the Western U.S.  We then examined potential differences in
simulated results as the atmospheric simulations are driven by meteorological fields
resolved with differing grid spacings and at different vertical levels.
We found that the observed average diurnal $CO_2$ pattern is better reproduced by
simulations driven by WRF-1.3km and WRF-4km ("AGL" configuration), with minimal
differences between the two configurations (Fig. 3).  The coarser-scale models (WRF-
12km_AGL, GDAS-1$^o$, and CarbonTracker) fail to reproduce the observed diurnal
pattern at all 3 sites.  The problem is especially severe at night, when both GDAS-ASL
and CarbonTracker lack the nocturnal enhancements.  In contrast, WRF-12km (AGL)
shows nocturnal $CO_2$ buildup that is clearly too strong.  The overestimation problem is
exacerbated when both coarser grid spacing and "AGL" configuration are adopted, as
seen in GDAS-AGL at HDP (Fig. 3).
The overestimate in nighttime $CO_2$ from WRF-12km (AGL) is due to the
preponderance of simulated air parcels found within the nocturnal PBL (Figs. 7~9),
which can be traced to the fact that air parcels are closer to the ground surface when
mountains are flattened.  Conversely, when released at "ASL" levels air parcels are found
much higher above the nocturnal PBL due to the flattening of mountains in a coarse-scale
global model like GDAS, resulting in minimal sensitivity to nighttime biospheric fluxes
and lack of $CO_2$ buildup.   Such large errors in estimated carbon fluxes due to lack of
ability to resolve patterns have also been found in earlier studies in Europe (Pillai et al.,
2011;Peters et al., 2010).
The natural question, then, is what can researchers do with mountaintop $CO_2$
observations, given the difficulty in resolving the terrain and flows in complex terrain?
**4.1  Approach 1:  Adjust vertical level of simulations from which to compare against**
**observed values**
The diurnal cycle simulated within CarbonTracker varies significantly as a function of
the vertical level (Fig. S5) from which $CO_2$ is extracted, particularly at night.  The
strongly attenuated diurnal cycle in the interpolated level corresponding to the ASL
elevation of the mountaintop sites (orange dashed) is found at higher levels within
CarbonTracker too, away from the first few levels near the ground.  At HDP, the



nighttime depletion of $CO_2$ at lower levels appears to be due to the erroneous nighttime
photosynthetic uptake in the gridcell where HDP is located (Fig. S2).
Interestingly, at SPL and NWR the diurnal pattern at a level between Levels 2 and 3
appears to correspond more closely to the overall observed $CO_2$ diurnal cycle, perhaps
due to the presence of nighttime enhancements closer to the model surface that is absent
from the higher levels closer to the ASL elevation.  The closer correspondence to
observed patterns may call for researchers to adjust the vertical level to maximize
resemblance to observations.  This was carried out at Jungfraujoch (Folini et al., 2008),
where the authors simulated carbon monoxide (CO) at multiple heights and arrived at a
height of 80 m above the model's ground surface as the best correspondence with the
observed CO, which was measured closer to the ground (Rinsland et al., 2000).  Instead,
a different study simulating observations at the same site adopted a height of 830 m
above the model ground surface (Tuzson et al., 2011).  This example illustrates the
divergence in researchers' choices for the vertical level in the midst of mountainous
terrain.
It is worth noting that the introduction of additional degrees of freedom in the vertical
level in "fitting" the measured $CO_2$ diurnal cycle within a carbon assimilation system is
potentially problematic.  The reason is that the assimilation system seeks to solve for
carbon fluxes by examining the mismatch between observed versus simulated $CO_2$
concentrations.  If the mismatch is due to erroneous fluxes, the introduction of additional
degrees of freedom in the vertical level would compensate for erroneous fluxes.  For
instance, if the nighttime carbon fluxes are overestimated in the model, this should show
up as an enhanced $CO_2$ concentration that is larger than observed values.  However, this
overestimation in $CO_2$ would be reduced by picking a higher vertical level rather than
fixing the overly large efflux in the model.  The optimal level could differ between night
and day as well; for instance, a level higher than Level 2 would fit better against
observations during the daytime at SPL and NWR (Fig. S5).  If different levels are
adopted at different times of the day, the degrees of freedom that can be adjusted would
be even larger, and model-data mismatches would be used in vertical level adjustments
instead of correcting erroneous biospheric fluxes.
Regardless, there is some role for vertical level adjustments to remove the gross
mismatch in the observed vs simulated diurnal cycles.  If the vertical level is indeed
adjusted in a carbon inversion system, we suggest that additional information (e.g.,
comparisons to meteorological observations or other tracers) is used rather than
maximizing the match to the target species (i.e., $CO_2$, in the case of a carbon inversion
system).
The $CO_2$ values at multiple levels within CarbonTracker show that unlike the
nighttime, differences between vertical levels are much smaller during the afternoon at
SPL and NWR (Fig. S5), suggesting that the simulated $CO_2$ values are not as sensitive to
the choice of vertical level.  We suspect that the large differences between vertical levels
at HDP is due to the flipped diurnal cycle in biospheric fluxes within CarbonTracker (Fig.
S3).  Otherwise, the lack of sensitivity to the choice of vertical level suggests that coarse-
scale models should assimilate afternoon observations, rather than nighttime observations
(see "Approach 4" below).



### 4.2 Approach 2: Reject mountaintop data

Due to the dangers of mis-representing terrain/flows and introducing biases into the carbon inversion system, an obvious way to deal with this problem is to neglect the mountaintop data altogether. This is already commonly practiced within carbon inversion systems (Rodenbeck, 2005;Geels et al., 2007;Peters et al., 2010). In fact, the most recent release of CarbonTracker ("CT-2015") stopped assimilating the three RACCOON sites (http://www.esrl.noaa.gov/gmd/ccgg/carbontracker/).

However, the absence of mountaintop $CO_2$ observations to constrain carbon inversion systems is, in effect, throwing away valuable information that could inform carbon exchange in potentially important areas of the world (Fig. 1). Case in point is the Schauinsland $CO_2$ time series on a mountain in the middle of Western Europe, which as of this writing has collected over 40 years of continuous $CO_2$ data (Schmidt et al., 2003) but remains excluded from numerous carbon inversion systems (Rodenbeck, 2005;Geels et al., 2007;Peters et al., 2010).

### 4.3 Approach 3: Assign errors to account for model errors

Instead of neglecting the mountaintop $CO_2$ observations altogether, an alternative approach is to make use of the observations, but assigning them errors within the model-measurement discrepancy error covariance matrix to account for model deficiencies (Lin and Gerbig, 2005;Gerbig et al., 2008). In this way, the inversion system would assign less weight to observations that the model has difficulties simulating. Given the systematic misrepresentation of the diurnal cycle in coarse-scale models, particularly at night (Fig. 3), this approach will effectively throw away much of the data as noise, due to inadequacies in the model. This naturally leads to the next possible approach of just having coarse-scale models assimilate afternoon observations.

### 4.4 Approach 4: Have coarse-scale models assimilate afternoon observations instead of nighttime

Our results show that the simulated $CO_2$ values are more in accordance with observed values in the afternoon (Fig. 3). This follows from the fact that afternoon trajectories and footprints match their higher resolution counterparts (Figs. 6, S7, S9, S10~S12), likely due to the deeper afternoon PBL depth and the reduction of terrain effects (Steyn et al., 2013). In other words, relative differences in PBL depth associated with flattening of mountains are lessened when the PBL is deeper; thus the impact on whether an air parcel sampled by the mountaintop site falls within the PBL is also attenuated under afternoon vigorous mixing conditions.

Based on these results, and in lieu of better transport, we suggest coarse-scale models may be better served to assimilate afternoon observations over the continent at their above sea level elevation. This is contrary to what has been commonly practiced by researchers, when nighttime mountaintop observations were assimilated (Peters et al., 2007;Keeling et al., 1976) to avoid daytime upslope flows and when nocturnal observations that represent free tropospheric conditions would better match coarse resolution models. We have found that sampling coarse-scale (1 deg) models at the corresponding ASL height have significant difficulties simulating nighttime $CO_2$, since it appears that the model failed to represent the strength of the nocturnal footprint at the 3 RACCOON mountaintop sites (Figs. 4, 5). Thus the inability of coarse-scale models to simulate the transport and PBL depths result in the lack of nocturnal enhancements and





thereby the wrong diurnal cycle (Fig. 3). Conversely, sampling the 12-km simulation at
the AGL height also has significant difficulties simulating nighttime $CO_2$, because it
overestimates the nocturnal footprint.
However, careful attention needs to be paid to upslope flows in the afternoon and
the potential mis-interpretation of more localized biospheric signals or anthropogenic
signals from below the mountain. A study from Jungfraujoch in Europe suggested that as
much as ~40% of the days in a year are influenced by thermally driven flows (Griffiths et
al., 2014). During the afternoon, the mountaintop site would then be influenced by
thermally driven upslope winds, as also pointed out by a number of studies around NWR,
along the Colorado Front Range (Sun et al., 2010;Sun and De Wekker, 2011;Parrish et al.,
1990) as well as SPL (De Wekker et al., 2009). For sites like HDP and NWR, which
have large nearby urban areas at lower elevation, upslope conditions can be of particular
concern if not properly accounted for. If these sites experience elevated $CO_2$ in the
afternoon from pollution sources, and this transport is not captured by the models, then
natural $CO_2$ sources can be significantly overestimated.
We found it encouraging that despite the proximity of significant population and
anthropogenic emissions from the Salt Lake and Denver area to the HDP and NWR sites,
respectively, the WRF-1.3km model suggests that the additional contribution of
anthropogenic $CO_2$ in the afternoon, over and beyond the nighttime signal is less than
1ppm, on average (Fig. S4). Presumably this is because of the high elevation of HDP and
NWR in relation to the urban area and the dilution of signals as they move up slope; the
afternoon urban signal would be enhanced if the sites were placed at lower peaks.
Regardless, it is prudent to consider mountaintop sites as not necessarily "pristine"
sites and to consider potential contributions from surrounding anthropogenic emissions
on these observations. It has been estimated that as of the year 2000, over 10% of the
world population live in mountainous areas (Huddleston et al., 2003), meaning that any
mountaintop site could very well see anthropogenic signatures. We recommend
additional tracers to be measured in conjunction with the mountaintop $CO_2$ sites. For
instance, combustion tracers such as $C^{14}$ and CO (Levin and Karstens, 2007) have been
measured alongside $CO_2$ at mountaintop sites in Europe. Another promising tracer is
$Rn^{222}$ (Griffiths et al., 2014), which provides a measure of surface exchange and would
help provide constraints on the exchange of air measured at the mountaintop with the
surface. Co-located meteorological observations—whether in-situ or remotely-sensed
(e.g., radar, sodar, lidar)—to probe atmospheric flows and turbulent mixing would also
be of significant value in helping to interpret the tracer observations (Rotach et al.,
2014;Banta et al., 2013).
**4.5 Approach 5: Adopt high-resolution modeling frameworks**
The least problematic, though potentially costly in terms of computational time, approach
to reduce modeling errors when interpreting mountaintop $CO_2$ observations is to adopt a
high resolution modeling framework. This conclusion was also arrived at by previous
studies (Pillai et al., 2011;van der Molen and Dolman, 2007;De Wekker et al., 2009).
From our results, it appears that meteorological fields from WRF at 4-km grid spacing,
driving a Lagrangian particle dispersion model, can reproduce most features from a 1.3-
km simulation, and generate a $CO_2$ diurnal cycle that qualitatively matches the observed
pattern. Once the WRF fields are degraded to 12-km grid spacing, the model fails to
capture such features.





While at least 4-km resolution in the meteorological fields is needed for the sites
examined here in the American Rockies, we anticipate that the minimum resolution
would depend on the level of complexity in the terrain, the height of the observational
site, and relationship with surrounding sources/sinks.
**5. Conclusions**
Given the large extent of the Earth's surface covered by hills and mountains and the large
amount of biomass and potential for carbon storage in complex terrain (Fig. 1), we call
for expanded efforts in observing and modeling $CO_2$ and other tracers on mountaintop
sites. This study has illustrated the potential for even coarse-scale models to extract
information from these observations when focusing on the daytime, afternoon values, and
the ability of high resolution models to simulate the general features of the summertime
diurnal $CO_2$ cycle even in the midst of significant terrain complexity. However, we
acknowledge that even the highest resolution model adopted in this paper undoubtedly is
subject to limitations of its own, and that deviations between simulated versus observed
$CO_2$ diurnal cycles arise from errors in both atmospheric transport as well as the
biospheric fluxes. Due to the focus on atmospheric transport in this paper, errors in the
simulations caused by shortcomings in the biospheric fluxes remain outside the scope of
this study (except for corrections to the flipped diurnal cycle; Fig. S3)
Even though current models remain imperfect, we call for sustained and expanded
observations of $CO_2$ and other tracers (e.g., CO, [222]Rn, and the isotopes of $CO_2$) co-
located with meteorological observations on mountaintop sites to create enhanced
datasets that can be further utilized by modeling frameworks of the future. Finally, we
call for testing and gathering of three-dimensional $CO_2$ observations over complex terrain,
as revealed by intensive airborne campaigns like the Airborne Carbon in the Mountains
Experiment (Sun et al., 2010).

**Acknowledgements**
This study was supported by NOAA Climate Program Office's AC4 program, award #
NA13OAR4310087 (UU). The RACCOON network has been supported by NSF (EAR-
0321918), NOAA (NA09OAR4310064) and DOE (DE-SC0010624 and DE-SC0010625).
The National Center for Atmospheric Research is sponsored by the National Science
Foundation. We thank site collaborators at the RACCOON sites: Ian and Gannet Hallar
at SPL, Dean Cardinale at HDP, and the CU Mountain Research Station and staff at
NWR. We thank J. Knowles for the meteorological data at NWR and S. Urbanski for
sharing the WFEI fire emission inventory. A. Andrews and A. Jacobson are gratefully
acknowledged for helpful discussions.



**Figure Captions**
**Fig. 1**
Aboveground biomass [mega-tons of carbon] from the North American Carbon Program
baseline dataset for year 2000 (Kellndorfer et al., 2013) overlaid on topographic surface
in the Western U.S., resolved at $0.5^o \times 0.5^o$ grid spacing.
**Fig. 2**
The WRF simulation domain, covering the Western U.S. with a series of nests with 12-,
4-, and 1.3-km grid spacing. These WRF meteorological fields are used to drive air
parcel trajectories within the STILT model.
**Fig. 3**
The average diurnal $CO_2$ pattern during June~August 2012 as observed at the 3
mountaintop sites in the RACCOON network: Hidden Peak (HDP), Storm Peak
Laboratory (SPL), and Niwot Ridge (NWR). Compared against the observations are
simulated diurnal $CO_2$ patterns from different models: CarbonTracker, STILT driven
with WRF at different grid spacings, and STILT driven with GDAS. Multiple GDAS-
driven STILT model configurations are shown, including runs without fixes to the
biospheric fluxes ("biofluxorig"; see Supplemental Information), as well as releasing air
parcels at the elevations of the sites above mean seal level ("ASL") or, for HDP only, at
the inlet height (Table 1) above the model's ground level ("AGL"). All of the WRF-
driven STILT runs place the release point of air parcels following the AGL configuration.
**Fig. 4**
The average diurnal footprint strengths at HDP, SPL, and NWR over June~August 2012
from STILT, driven with different meteorological fields and release heights (ASL vs
AGL). The footprint strength was derived by summing over the spatial distribution of
footprint values (Fig. 5).
**Fig. 5**
The average footprint (shown in $log_{10}$) for the Hidden Peak (HDP) site in Utah, at night:
0200 MST (0900 UTC), gridded at $0.1^o \times 0.1^o$. The site is denoted as a triangle. The
average back trajectory (averaged over the stochastic STILT trajectories) is drawn as a
line, with points indicating trajectory locations every hour, as the trajectory moves back
from the site indicated as points. Red parts of the trajectory refer to the nighttime
(1900~0700 MST), while pink portions indicate the daytime (0700~1900 MST). Parts of
the trajectory are shaded with blue when it is found below the average height of the PBL
along the trajectory.
**Fig. 6**
Similar to Fig. 5, but for the afternoon: 1400 MST (2100 UTC) at HDP.
**Fig. 7**
Three dimensional plots of the terrain over a domain of $\sim 1^o \times 1^o$ surrounding HDP, as
resolved by the WRF and GDAS models at various grid spacings. The HDP site is
denoted as a triangle. Also shown is the average back trajectory, derived by averaging
locations of the numerous stochastic trajectories simulated by STILT, driven by the





various WRF meteorological fields and the global GDAS field. Back trajectories were
started from HDP at 0200 MST (0900 UTC). Points indicate trajectory locations every
hour, as the trajectory moves back from the site indicated as points. Red portions of the
trajectory refer to the nighttime (1900~0700 MST), while pink portions indicate the
daytime (0700~1900 MST). In addition, the PBL heights averaged along the
backtrajectory are shown as the blue line.
**Fig. 8**
Time series of the average back trajectory and PBL heights relative to the ground surface
("AGL") instead of above sea level, at each time step backward in time from the receptor
(triangle). Red portions of the trajectory refer to the nighttime (1900~0700 MST), while
pink portions indicate the daytime (0700~1900 MST). The PBL heights averaged along
the backtrajectory are shown as the blue line. The nighttime PBL height is indicated in
dark blue, while the daytime portion is in light blue.
**Fig. 9**
Similar to Fig. 7, but for the Storm Peak Laboratory (SPL) site.
**Fig. 10**
Similar to Fig. 8, but for the Storm Peak Laboratory (SPL) site.
**Fig. 11**
Similar to Fig. 9, but for the Niwot Ridge (NWR) site.
**Fig. 12**
Similar to Fig. 10, but for the Niwot Ridge (NWR) site.
**Fig. 13**
Similar to three-dimensional terrain and trajectory plots as shown in Figs. 7, 9, and 11,
but for just the GDAS 1 deg. ASL simulations and for the morning hours of 0800 MST
and 1100 MST.



**Tables**

|  | Hidden Peak (HDP) | Storm Peak Lab (SPL) | Niwot Ridge (NWR) |
|---|---|---|---|
| Latitude/Longitude | 40º 33' 38.80" N 111º 38' 43.48" W | 40º 27' 00" N 106º 43' 48" W | 40º 03' 11" N 105º 35' 11" W |
|  |  |  |  |
| Top Inlet Height | 17.7 m | 9.1 m | 5.1 m |
|  |  |  |  |
| Site Altitude [m above sea level] | 3351 m | 3210 m | 3523 m |
| Model Altitude: |  |  |  |
| *WRF-1.3km* | 2996 m | 3038 m | 3411 m |
| *WRF-4km* | 2918 m | 2818 m | 3382 m |
| *WRF-12km* | 2357 m | 2724 m | 3076 m |
| *GDAS* | 1856 m | 2757 m | 2333 m |
| *CarbonTracker* | 2004 m | 2582 m | 2276 m |

**Table 1**. Characteristics of RACCOON mountaintop sites examined in this paper, as
well as the representation of terrain in different meteorological files at these sites.






| Site | SPL | | | | NWR | | | |
|---|---|---|---|---|---|---|---|---|
| Run type | 1.3-km WRF | 4-km WRF | 12-km WRF | GDAS | 1.3-km WRF | 4-km WRF | 12-km WRF | GDAS |
| u-wind BIAS [m/s] | -0.5 | -1.5 | -0.9 | 2.3 | 0.1 | -0.3 | -1.4 | -0.2 |
| v-wind BIAS [m/s] | -0.6 | -0.3 | -0.2 | 1 | 0.2 | 0.4 | 0.9 | 1.1 |
| u-wind RMSE [m/s] | 3.1 | 3.8 | 3.2 | 3.7 | 3.5 | 3.4 | 3.4 | 3.2 |
| v-wind RMSE [m/s] | 2.7 | 2.7 | 2.3 | 2.5 | 2.2 | 2.1 | 2.2 | 3 |

**Table 2**. Comparisons of different meteorological files driving STILT against hourly-
averaged wind observations at Storm Peak Laboratory (-106.74 W; 40.45 N) and at
Niwot Ridge (-105.586 W; 40.053 N; 3502 m ASL) (Knowles, 2015), near the
RACCOON $CO_2$ site. Meteorological observations were not available at the Hidden Peak
site. Error statistics are presented separately for the west-to-east component ("u-wind")
and south-to-north component ("v-wind") of the wind velocity vector.





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

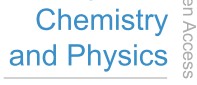
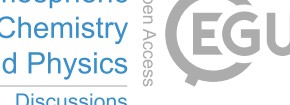


## Above-ground Biomass in the Western U.S.

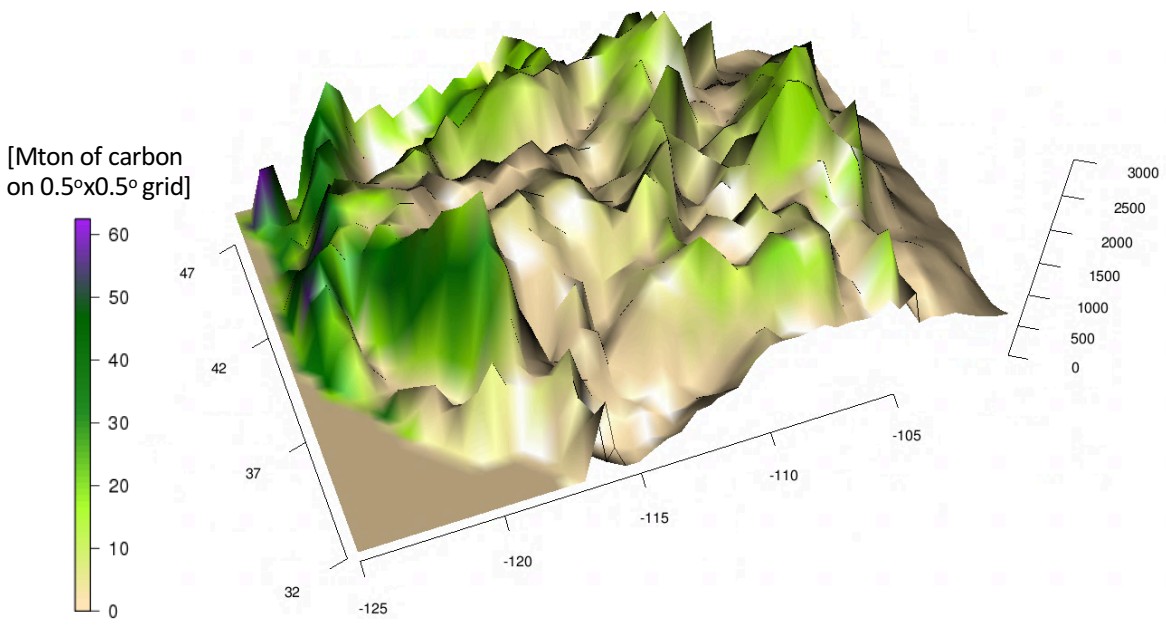

**Fig. 1**





# WRF Domains

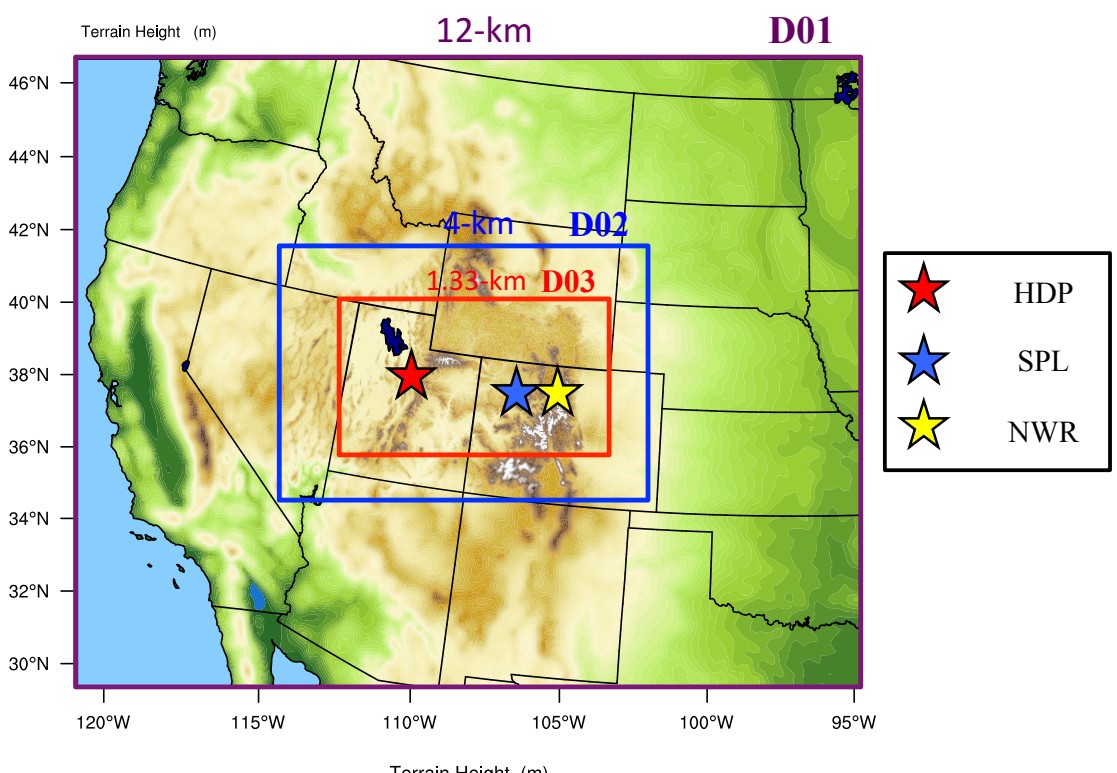

**Fig. 2**



**Fig. 3**







**Fig. 4**





**HDP ave footprint:  WRF-1.3km (AGL)**

**HDP ave footprint:  WRF-4km (AGL)**

**0900 UTC
(0200 MST)**

**Nighttime
(1900~0700 MST)**

**Daytime
(0700~1900 MST)**

**Trajectory in
PBL**

**HDP ave footprint:  WRF-12km (AGL)**

**HDP ave footprint:  GDAS-1° (ASL)**

**Fig. 5**





**HDP ave footprint:  WRF-1.3km (AGL)**

**HDP ave footprint:  WRF-4km (AGL)**

**2100 UTC
(1400 MST)**

Nighttime
(1900~0700 MST)

Daytime
(0700~1900 MST)

Trajectory in
PBL

**HDP ave footprint:  WRF-12km (AGL)**

**HDP ave footprint:  GDAS-1° (ASL)**

**Fig. 6**



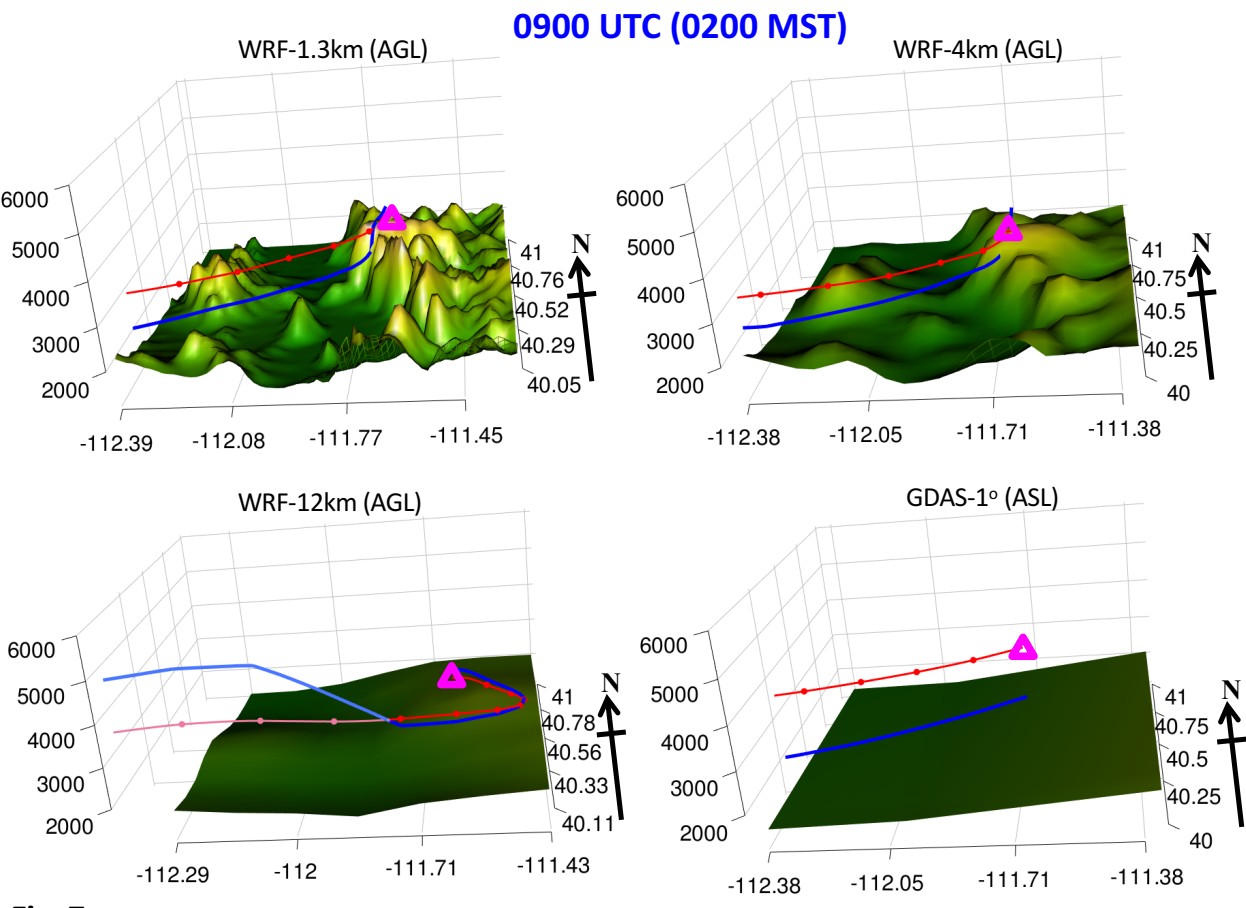

**Fig. 7**





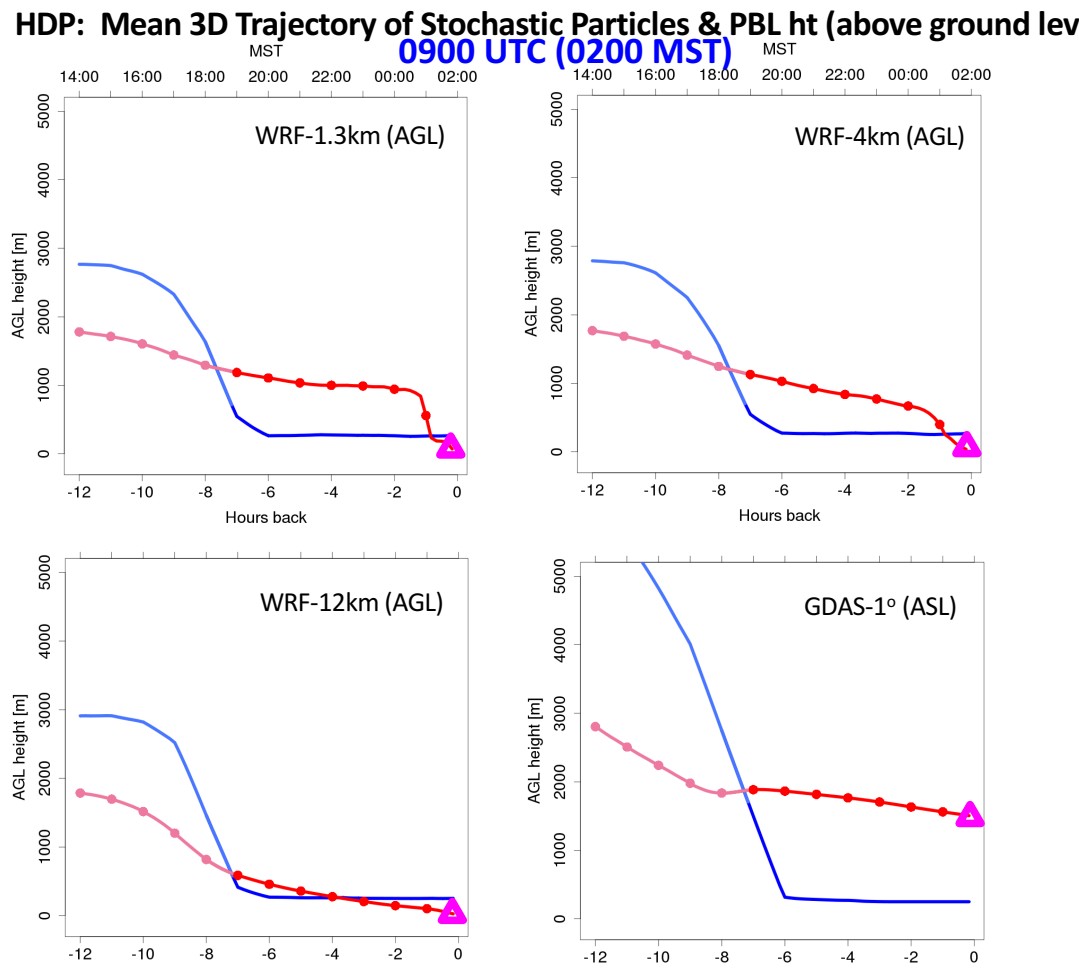

Fig. 8



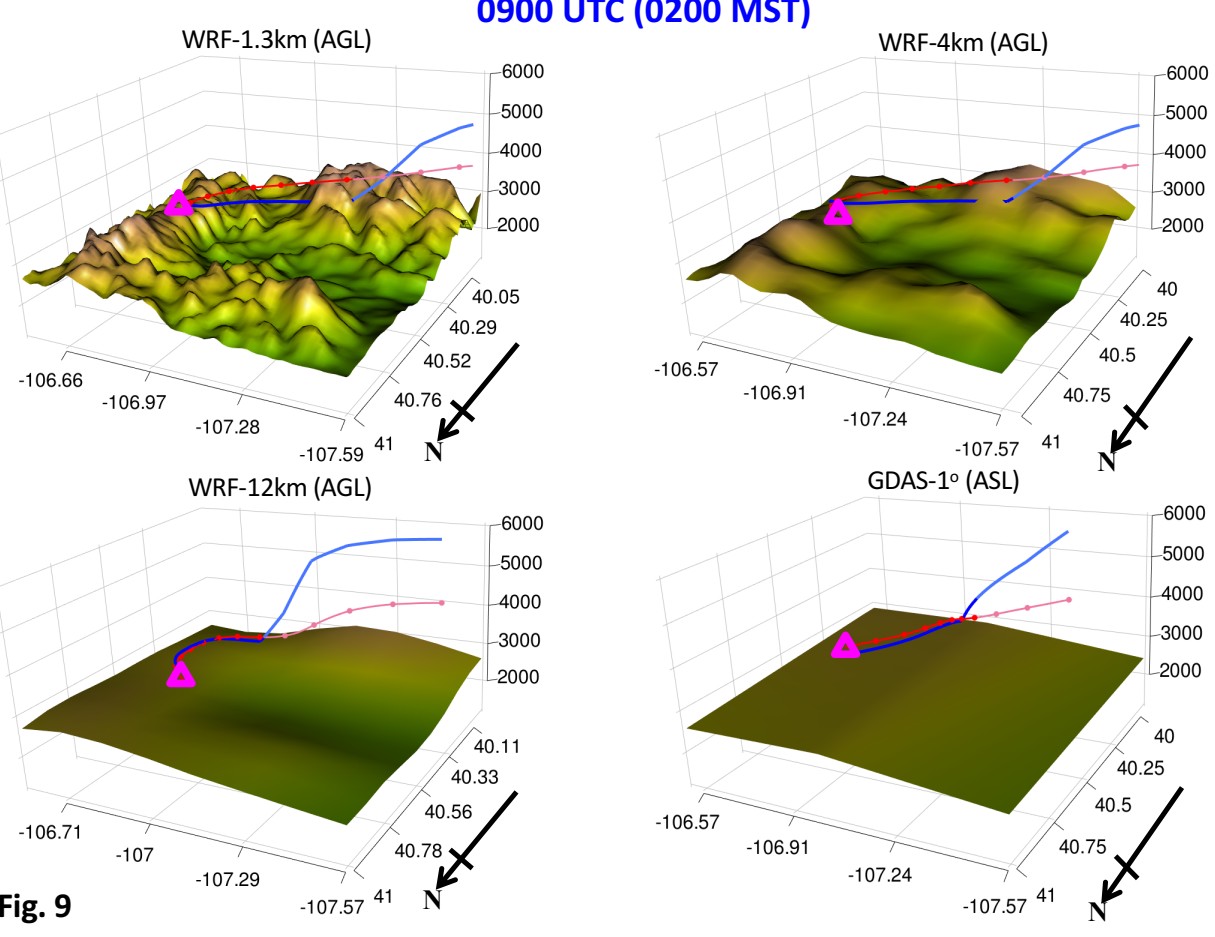

**Fig. 9**



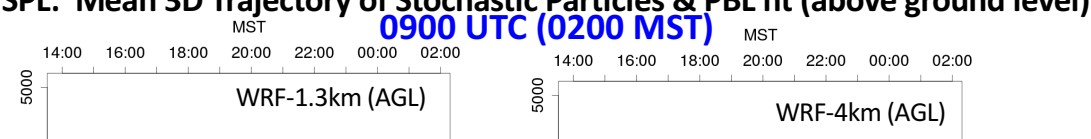

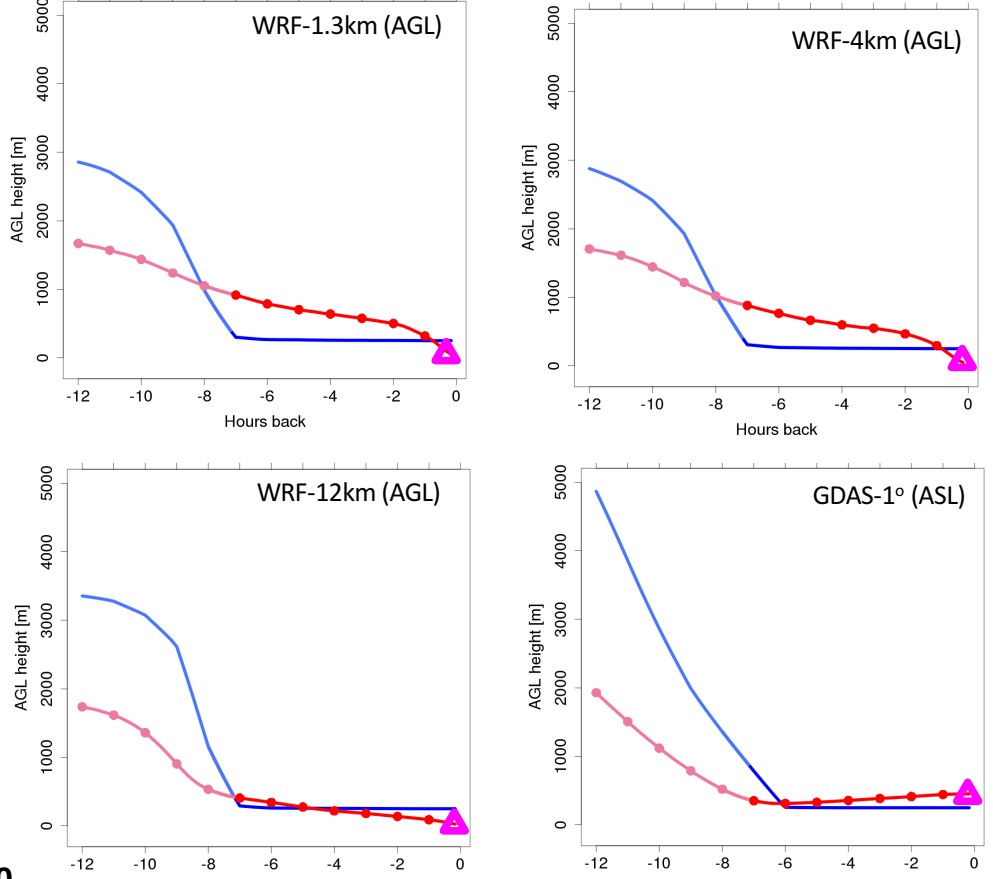

**Fig. 10**



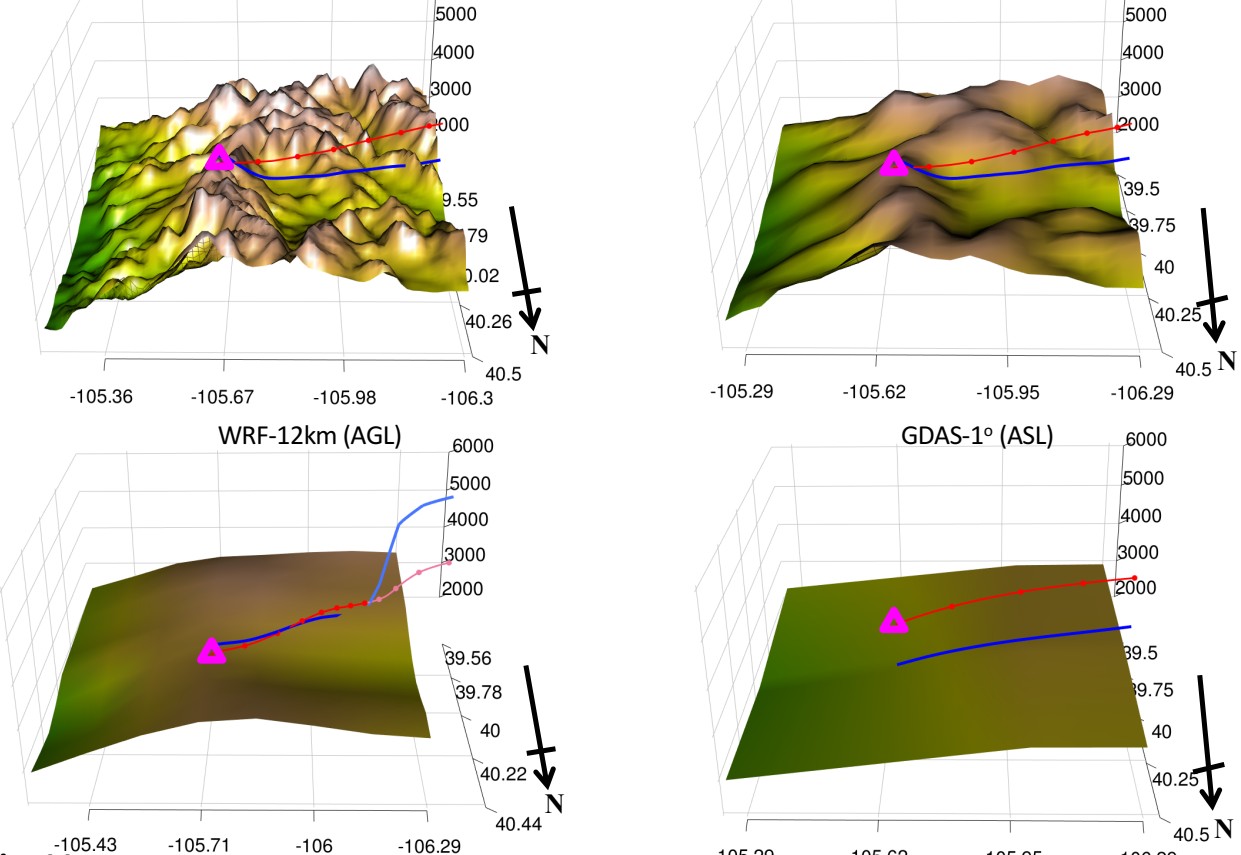

**Fig. 11**



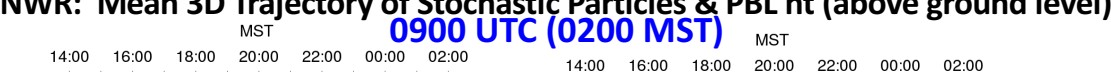

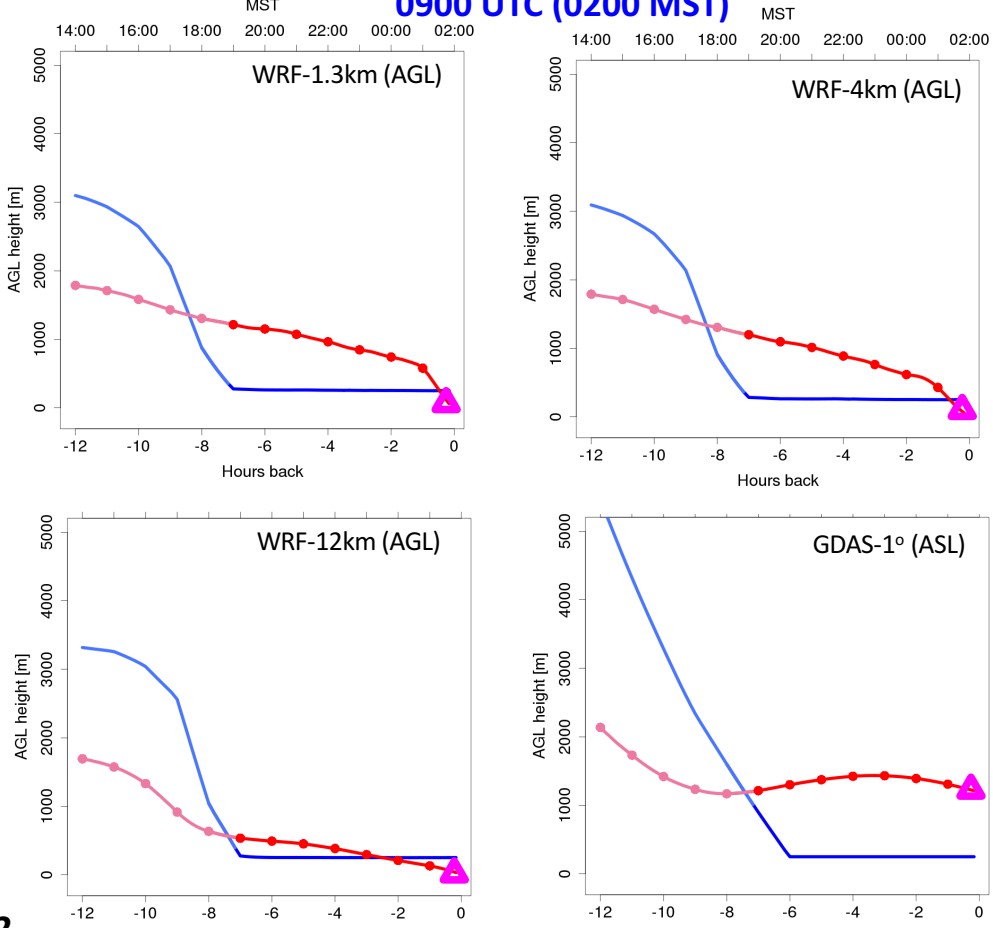

**Fig. 12**



**Mean 3D Trajectory of Stochastic Particles & PBL ht**
**GDAS-1° (ASL)**

**Fig. 13**