# Peer review of "How can mountaintop $CO_2$ observations be used to constrain regional carbon fluxes?"

_Atmospheric Chemistry and Physics, 2016_

## Referee Comment (RC1) · Anonymous Referee #1 · 8 Dec 2016

This study examined the ability of different atmospheric transport models (and with different spatial resolutions) to simulate CO2 mixing ratios (specifically the summer time diurnal cycle) at mountaintop locations in western USA. The results showed that regional transport model (Lagrangian Particle Dispersion Model STILT) with higher resolution (WRF at 4 km or less) wind inputs compared better with observations than lower resolution models (WRF at 12 km or CarbonTracker). The comparison and analysis with global model (CarbonTracker) point to some approaches in using mountaintop observations in carbon flux estimates. This type of model evaluation is important to understand the abilities and shortcomings of transport models, and necessary prior to using such models in flux estimation. The results provided more insights on the importance of local scale up slope and down slope winds for mountain top observations. Therefore this paper should be accepted for publication following some minor

modifications noted below.

General Comments

The WRF-STILT model results at 4 and 1.3 km resolutions compared well with observation. But the differences between WRF-STILT model results and observations became unexpected large at 12 km resolution (Fig. 3), with poorer agreement than the much lower resolution models (GDAS-STILT and CarbonTracker, both at 1 degree resolution). The problem seems to be the modeled PBL. The 12 km WRF-STILT model mean PBL shown in Figs. 7, 8 and 9 are quite different from the WRF-1.3, WRF-4 and GDAS-1 deg results. Similarly the GDAS-STILT results exhibit odd behavior (see below). This raises many questions:

(1) Is there a problem with the modeling of PBL in the WRF-12km model?

(2) Is there a problem with the modeling of PBL in the GDAS-STILT model (Figs. 10, S10, S12)? Figs. S10 and S12 show HDP and NWR PBL much higher (~5000m) compared to PBL from WRF-STILT runs. Fig. 10 appears mislabeled and not consistent with the discussion in the text, therefore very difficult to understand, see specific comment.

(3) How realistic are the PBL results in these models (WRF-1.3km, WRF-4km, WRF-12km, GDAS)? How were the PBL results evaluated (for mountaintop conditions)?

(4) How realistic is PBL model for mountaintop simulations? How does the PBL model account for the difference between flat plain and mountainous conditions (as the model domain contains such surface conditions)?

The authors should provide more technical details on the PBL model (e.g. dependence on atmospheric conditions and topographic variations, daytime and nighttime variations), particularly how it changes for the different (met data) models (to cause the unusual results for WRF-12km and GDAS) in this study. Then present more analysis and discussion on the uncertainties or errors in the PBL and consequently footprints in

this study.

Specific comments

Lines 413-414: 'Due to the significantly "flattened" mountains in WRF-12km and in GDAS, the PBL height exhibits less spatial variation.' This is not true for WRF-12km in Figure 7.

Lines 469-472: 'Focusing on the three-dimensional plots at the hours of 0800 and 1100 MST (Fig. 10), when the simulated peaks are found at SPL and both NWR/HDP, respectively, the peaks coincide with times when average trajectories are found within a relatively shallow morning PBL.' This statement seems to conflict with Fig. 10, which shows the HDP site with a deep morning average PBL. There seems to be error(s) in Fig. 10. The HDP plot domain is the same as shown for NWR in Fig. 9. The plot domain for NWR is as for HDP in Fig. 7. The plotted curves for the two upper plots with nearby domains seem to show drastic difference for PBL, the PBL for the HDP plot is definitely not 'shallow morning PBL' at ∼6000m.

Lines 734-736: 'Red portions of the trajectory refer to the nighttime (1900∼0700 MST), while pink portions indicate the daytime (0700∼1900 MST).' The red and pink portions are hard to distinguish, use more contrasting colors.

Lines 738-739: 'Fig. 8 Similar to Fig. 7, but for the Storm Peak Laboratory (SPL) site.' The blue PBL line appears to have dark and light portions not explained. Again, the contrast is hard to distinguish, use more contrasting colors. This may apply to Figs. 7, 9, 10 too.

Figure 7 (also 8, 9, 10, and corresponding figures in supplemental material): add color scale to facilitate figure comparisons.

Supplemental material

Figure S3: it is not clear if the flux reversal correction takes into account the longer daytime than nighttime during the summer.

Figure S5: 'CO.obs...' to 'CO2.obs...'

Figs. S11, S12: PBL for GDAS higher than 6000m seems unusually high.

---

## Referee Comment (RC2) · Anonymous Referee #3 · 15 Jan 2017

Comments on "How can mountaintop CO2 observations be used to constrain regional carbon fluxes?" by Lin et al.

Overall comments: This study intends to investigate the mountaintop CO2 observations being used to constrain the carbon fluxes in regional scales. The authors compared the CO2 simulations and trajectories with CO2 observations at 3 sites in the mountains of the Western U.S. in summer from June to August of 2012. The authors also have adopted multiple approaches to discuss and address possible contributors of the results from STILT coupled with individual meteorological model outputs or dataset. They indicated that a fine grid spacing of ∼4 km or less may be needed to simulate a realistic diurnal cycle of CO2 for sites on top of the steep mountains for avoiding erroneous atmospheric flows as a result of terrain that is misrepresented in the model.

The major concern is that how different the CO2 simulations and observations are compared from day to day, particularly when finer scale models are used as suggested? Averaged results might miss some crucial information hidden in the discrepancies between model simulations and site observations in present study.

The related studies focusing on mountaintop such as this one are important and rarely explored. I would recommend this manuscript for publication after above concern is addressed.

Specific comments: Line 146: How is the vegetation covered at these mountain stations? Are these monitoring sites far above the tree line? Local influences on CO2 due to the surrounding plant cover at these sites may need to be considered.

Lines 296-297: How much percent is contributed to the wildfires during the study period?

Lines 301-302 and Fig. 3: Fig. 3 illustrates the 3-month averaged diurnal cycles of the results from individual models and observations. Do they show similar diurnal cycles every day during these months? Consider addressing the variabilities of the diurnal cycles (e.g. error bars) in the supplement.

Fig. 4: Has the GDAS_1-deg_ASL in Fig. 4 been adjusted with the biospheric fluxes? If not, I would recommend drawing it as gray line and dot as presented in Fig. 3.

Fig. S5: Two CT Level 1 (dark blue) and two CT Level 2 (blue)?

Fig. S5 and Table 1: The differences between the site and model altitude (Table 1) seem to be generally associated with the CO2 biases. For instance, larger differences such as WRF-12km, GDAS, and Carbon Tracker showed larger discrepancies in diurnal cycles of CO2 at HDP. What if the model altitude fixed as same as the mountain site? Could it be better correlated to the observational CO2 data?

Lines 336-380: Are these trajectories in a good agreement for each day during the study period both in nighttime and afternoon hours?

Please also note the supplement to this comment:
http://www.atmos-chem-phys-discuss.net/acp-2016-762/acp-2016-762-RC2-
supplement.pdf

---

## Referee Comment (RC3) · Anonymous Referee #2 · 16 Jan 2017

Summary

Lin et al. investigate the modelled CO2 concentration time series for June to August 2012 at 3 atmospheric measurements stations with RACCON observations. Carbon fluxes from established inventories and models are used as input for different atmospheric transport model configurations (x,y = 1.3km, 5km, 12km and 1degree) to investigate the impact of their resolution (especially the ability to reproduce the topography of the domain) on the model-data mismatch. Mean daily cycles of CO2 concentrations are analysed as well as the difference in the modelled back-trajectories. The authors results motivate 5 different approaches how to use (or not use) such observations in future studies.

General comments: The paper is well-written and the observational datasets, models

and methods are properly described and referenced. The authors address an urgent question about the future and current use of observations from mountainous areas or within complex terrains more general. Their analysis of mean daily cycle allows an easy interpretation of mean behaviour, but it would be critical to also investigate if there are episodes (meteorological conditions) for which the mismatch between the different transport model configuration is minimized/maximized. I would suggest to add an analysis of the time-dependant offset of each model compared to the observations. In general it would be valuable to have more quantitative results and a discussion how generalizable the findings of this study are for other hilly/mountainous areas. Overall this paper also does not fully address the question of "constraining carbon fluxes", but rather how well different model setups can reproduce the atmospheric concentrations of CO2. To be able to really judge if the models (even the best, 1.3km resolution) are able to e.g. distinguish different prior carbon flux estimates the authors would need to perform a sensitivity study using multiple carbon flux data sets and demonstrate a significant impact at the three sites. Potentially one can consider this study rather a step towards a better use of such data rather than already addressing the question of regional carbon fluxes. After the above and below comments are addressed I would definitely recommend this study for publication in ACP as it will help to better understand limitations of such observations and were future model developments should/could be focussed to eventually be able to constrain carbon fluxes in such regions.

Specific comments: Line 44ff: The claim that nearly 70% of the earth land surface is covered by hills or mountains needs to be better validated. This surely depends on the definition for hill or mountain, which is not given here and the cited publication is hard to access (due to the journal it was published in) and the journal has an impact factor below 1. The authors also mention that carbon fluxes in complex terrain need to be better understood to quantify carbon flux. It seems you are implying that all mountainous or hilly areas are (too) hard to model? Line 193: The authors refer to a previous publication – nonetheless the key parameters e.g. vertical mixing scheme used should be explicitly given in this publication (e.g. by adding a table in this section).

[Figure]

Line 216: Please specify if the system allows for two-way nesting or not Line 277: A "fix" is mentioned, but not explained at all. Please consider giving a brief description here rather than referring to the supplement. It seems the daily cycle has just been shifted or were there any more complicated adjustments performed? Line 304: Please consider refering to table 1 here so the reader can easily find the height CT data was extracted from. Line 503ff: The first question is repeated here "How can mountaintop $CO_2$ observations be used to constrain regional scale carbon fluxes, ...." But the 5 subsections following rather discuss IF such data can be used or how they can be better used. It remains unclear if there is a definitive answer on how to use them. Line 530ff: Choosing the appropriate model layer to extract $CO_2$ does indeed introduce a significant additional degree of freedom. The authors suggest other parameters to avoid creating a fudge factor but do not give specific advice here on which tracers could be useful ($^{222}Rn$?). Meteorological data is mentioned but looking at table 2 it seems not at all clear that this would be good parameter or what a cut-off would be. Could you suggest how the a suitable proxy could be found? Line 581ff: Here the authors report on the practice of not using mountaintop data but it is unclear how this is linked to this specific study as no suggestion is made how to e.g. better use Schauinsland data. This section should be considered for the introduction to motivate why Approach 1, 3, 4, 5 need to be improved. Line 595ff: When setting up an inversion system it is common (good) practice to assign proper model errors. This seems not specific to this study and the authors fail to give an estimate of the model error for the three sites discussed here. Please consider removing this section or giving quantitative results for the sites and models investigated here. Of course, the model data mismatch calculated here also depends on flux errors, but the authors can surely use this study to give an upper limit of this combined error (and the difference for different model resolutions).
* * *

---

## Author Response (AR1)

We thank Reviewer #1 for the constructive criticism of the Discussion paper. The reviewer's comments are shown below in *italics*, while our point-by-point responses are indicated as un-italicized.

**Reviewer #1**

*General Comments*

*The WRF-STILT model results at 4 and 1.3 km resolutions compared well with observation. But the differences between WRF-STILT model results and observations became unexpected large at 12 km resolution (Fig. 3), with poorer agreement than the much lower resolution models (GDAS-STILT and CarbonTracker, both at 1 degree resolution). The problem seems to be the modeled PBL. The 12 km WRF-STILT model mean PBL shown in Figs. 7, 8 and 9 are quite different from the WRF-1.3, WRF-4 and GDAS-1 deg results. Similarly the GDAS-STILT results exhibit odd behavior (see below).*

The results from the modeled PBL scheme actually do not differ appreciably in the three WRF configurations. This was made much clearer by our addition of new figures for the Discussion paper (Figs. 8, 10, 12) showing the time series of PBL heights (and trajectories) with respect to above-ground level (AGL). The key difference is the smoothing of the terrain in the coarser simulations. We have added new text to clarify this point:

"         An alternative perspective is to view the trajectory and PBL heights relative to the ground surface ("AGL") instead of above sea level, at each time step backward in time from the receptor (Figs. 8, 10, 12). These figures highlight the fact that while PBL dynamics in the three WRF configurations are similar, the heights of the trajectories relative to the PBL height differ.   The trajectory exits above the nocturnal PBL one hour backward in time, on average, while the WRF-12km trajectory spends several hours within the PBL.

         The difference in the trajectory behavior can be explained by the differing terrain. In mountainous terrain, PBL heights generally follow the terrain elevations, albeit with attenuated amplitude (Steyn et al., 2013). Thus in WRF-1.3km and 4km, the more highly resolved terrain produced shallow nocturnal PBL height that descend in the valley (Fig. 7) while the corresponding trajectory hovers above it. Viewed relative to the ground surface (Fig. 8), the trajectory originating from HDP appears to have exited above the nocturnal PBL in WRF-1.3km and 4km. In contrast, due to the significantly "flattened" mountains in WRF-12km and in GDAS, the PBL heights exhibit less spatial variation near the mountaintop receptor, since the terrain itself was smoothed. Consequently, WRF-12km trajectories, unlike the WRF-1.3km or -4km cases, travel closer to the ground surface, within the nighttime PBL, even as it is advected away from the three RACCOON sites (Figs. 7, 8). This resulted in stronger nighttime footprints in WRF-12km as seen in Figs. 4 and 5. Another effect of the proximity of the air parcels to the model's ground surface is the slower windspeeds from surface drag, causing the air parcel trajectories to remain close to the 3 sites until the previous day; for HDP and SPL, the mean trajectories spiral toward the site at the surface, following an "Ekman wind spiral" pattern (Holton, 1992). In WRF-1.3km and WRF-4km, the measurement sites are at significantly higher elevations above the resolved valleys in the area surrounding the sites, and the air parcels are found above the shallow nocturnal boundary layer hugging the valley floor, on average (Fig. 7)."

*This raises many questions:*
*(1) Is there a problem with the modeling of PBL in the WRF-12km model?*
No--please see response above.

*(2) Is there a problem with the modeling of PBL in the GDAS-STILT model (Figs. 10, S10, S12)? Figs. S10 and S12 show HDP and NWR PBL much higher (5000m) compared to PBL from WRF-STILT runs.*
We believe the considerable difference in behavior within GDAS is due to both the significantly coarser vertical and horizontal resolutions.

We now explain the impact of this coarse resolution:

"Another noticeable difference in GDAS-ASL trajectory was the significantly higher daytime PBL heights (Figs. 8, 10, 12).  We suspect this is because of the greatly reduced vertical resolution within GDAS (23 levels versus 41 levels in WRF):  since STILT diagnoses the PBL height to correspond to a model level, a higher PBL height was chosen for GDAS because of the thicker vertical level.  Another subtle artifact of the coarse resolution within GDAS can be seen in the anomalously low daytime PBL height just in the vicinity of HDP (Figs. 13, S10).  It appears that the GDAS model set an entire $1^{o} \times 1^{o}$ grid box near HDP to be water body (the Great Salt Lake), thereby suppressing the PBL height."

*Fig. 10 appears mislabeled and not consistent with the discussion in the text, therefore very difficult to understand, see specific comment.*
We believe that the impression of Fig. 10 being mislabeled likely resulted from the Reviewer looking at an older version of the manuscript instead of the version published as the Discussion paper. See below ("Specific Comments") for details.

*(3) How realistic are the PBL results in these models (WRF-1.3km, WRF-4km, WRF-12km, GDAS)? How were the PBL results evaluated (for mountaintop conditions)?*
*(4) How realistic is PBL model for mountaintop simulations? How does the PBL model account for the difference between flat plain and mountainous conditions (as the model domain contains such surface conditions)?*

*The authors should provide more technical details on the PBL model (e.g. dependence on atmospheric conditions and topographic variations, daytime and nighttime variations), particularly how it changes for the different (met data) models (to cause the unusual results for WRF-12km and GDAS) in this study. Then present more analysis and discussion on the uncertainties or errors in the PBL and consequently footprints in this study.*

The aforementioned questions (3) and (4) are linked and currently difficult to answer. How to carry out PBL simulations in mountainous terrain and how to evaluate them very much remains unsettled.  The requisite meteorological and tracer observations (in addition to $CO_2$) are limited for the sites examined in this paper. We fully recognize this difficulty and have already included a recommendation for additional observations in future mountaintop sites:

" We recommend additional tracers to be measured in conjunction with the mountaintop $CO_2$ sites. For instance, combustion tracers such as $C^{14}$ and CO (Levin and Karstens, 2007) have been measured alongside $CO_2$ at mountaintop sites in Europe. Another promising tracer is $Rn^{222}$ (Griffiths et al., 2014), which provides a measure of surface exchange and would help provide constraints on the exchange of air measured at the mountaintop with the surface. Co-located meteorological observations—whether in-situ or remotely-sensed (e.g., radar, sodar, lidar)—to probe atmospheric flows and turbulent mixing would also be of significant value in helping to interpret the tracer observations (Rotach et al., 2014;Banta et al., 2013)."

*Specific comments*
*Lines 413-414: 'Due to the significantly "flattened" mountains in WRF-12km and in GDAS, the PBL height exhibits less spatial variation.' This is not true for WRF-12km in Figure 7.*
This actually still holds for WRF-12km in Fig. 7, but is true, strictly speaking, only for the key area near the receptor. We have now clarified this point to:

" In contrast, due to the significantly "flattened" mountains in WRF-12km and in GDAS, the PBL heights exhibit less spatial variation near the mountaintop receptor, since the terrain itself was smoothed."

*Lines 469-472: 'Focusing on the three-dimensional plots at the hours of 0800 and 1100 MST (Fig. 10), when the simulated peaks are found at SPL and both NWR/HDP, respectively, the peaks coincide with times when average trajectories are found within a relatively shallow morning PBL.' This statement seems to conflict with Fig. 10, which shows the HDP site with a deep morning average PBL. There seems to be error(s) in Fig. 10. The HDP plot domain is the same as shown for NWR in Fig. 9. The plot domain for NWR is as for HDP in Fig. 7. The plotted curves for the two upper plots with nearby domains seem to show drastic difference for PBL, the PBL for the HDP plot is definitely not 'shallow morning PBL' at _6000m.*
Reviewer #1 appears to be looking at an earlier version of paper and not the version published as the Discussion paper. Lines 469-472 have different text now, and we believe the text in the published Discussion paper matches what is shown in the figures.

*Lines 734-736: 'Red portions of the trajectory refer to the nighttime (1900_0700 MST), while pink portions indicate the daytime (0700_1900 MST).' The red and pink portions are hard to distinguish, use more contrasting colors.*
We thank the Reviewer for pointing this out. We have followed the suggestion and changed the colors to show more contrast.

*Lines 738-739: 'Fig. 8 Similar to Fig. 7, but for the Storm Peak Laboratory (SPL) site.' The blue PBL line appears to have dark and light portions not explained. Again, the*

*contrast is hard to distinguish, use more contrasting colors. This may apply to Figs. 7, 9, 10 too.*
We thank the Reviewer for pointing this out. We have followed the suggestion and changed the colors to show more contrast.

*Figure 7 (also 8, 9, 10, and corresponding figures in supplemental material): add color scale to facilitate figure comparisons.*
We do not feel that the color scale is necessary, since the colors represent elevation and the same information is also represented by the z-axis, as well as the three-dimensional terrain representation. Furthermore, these figures already incorporate a lot of information, so adding another colorscale could make the figures look too "busy".

*Supplemental material*

*Figure S3: it is not clear if the flux reversal correction takes into account the longer daytime than nighttime during the summer.*
Yes--the longer daytime can be accommodated in our flux reversal correction algorithm. The algorithm preserves the area between the flux time series and the flux = 0 line. So to the extent that the diurnal pattern in the uncorrected CarbonTracker fluxes shows a longer period of release or uptake, this would show up in the corrected fluxes too. In other words, the algorithm preserves the relative length of carbon uptake or release in the original CarbonTracker fluxes, but any nighttime uptake would be moved to the daytime.

*Figure S5: 'CO.obs...' to 'CO2.obs...'*
We thank the reviewer for pointing out this error. This figure has now been fixed.

*Figs. S11, S12: PBL for GDAS higher than 6000m seems unusually high.*
We believe the high PBL within GDAS is mainly due to the coarse vertical grid spacing. We have added an explanation of this point:

"Another noticeable difference in GDAS-ASL trajectory was the significantly higher daytime PBL heights (Figs. 8, 10, 12). We suspect this is because of the greatly reduced vertical resolution within GDAS (23 levels versus 41 levels in WRF): since STILT diagnoses the PBL height to correspond to a model level, a higher PBL height was chosen for GDAS because of the thicker vertical level."

We thank Reviewer #2 for the constructive criticism of the Discussion paper. The reviewer's comments are shown below in *italics*, while our point-by-point responses are indicated as un-italicized.

**Reviewer #2**

*General comments:*

*The paper is well-written and the observational datasets, models and methods are properly described and referenced. The authors address an urgent question about the future and current use of observations from mountainous areas or within complex terrains more general. Their analysis of mean daily cycle allows an easy interpretation of mean behaviour, but it would be critical to also investigate if there are episodes (meteorological conditions) for which the mismatch between the different transport model configuration is minimized/maximized. I would suggest to add an analysis of the time-dependent offset of each model compared to the observations.*

We agree with the Reviewer that beyond the diurnal timescale, there is day-to-day variability in the model behavior that lead to variations in $CO_2$ model errors. We have now included in the revised paper the time series of $CO_2$ model errors at SPL, NWR and HDP (see below) with correlations of the $CO_2$ errors with various meteorological variables (geopotential height and its E⇔W gradient, U- and V-winds, and the windspeed). The time series plots and a Table of the correlations of the $CO_2$ errors with these meteorological variables will be added to the Supplement.

Errors at multi-day timescales are the most strongly correlated with different meteorological variables, depending on which is being examined: V-wind, U-wind, and geopotential height gradient for HDP, SPL, and NWR, respectively. The fact that errors are correlated with different meteorological variables depending on the site location points to a complexity that can only be unraveled with a substantial expansion of the paper. This could potentially be a subject for a future paper. This complexity is in contrast to the average diurnal biases that can in large part be linked to the underlying resolution of modeled terrain, which the current paper focuses on.

[Figure]

[Figure]

*In general it would be valuable to have more quantitative results and a discussion how generalizable the findings of this study are for other hilly/mountainous areas.*
We do attempt to broaden the scope of this study. In particular, we outline various approaches in the Discussion section with regards to making use of mountaintop $CO_2$ data that can be considered for other mountainous areas as well. However, we believe detailed quantitative results for other mountainous areas would require dedicated modeling efforts. This need for dedicated efforts for individual mountainous sites is hinted at by differences in results between the three sites examined in this study: HDP, SPL, and NWR (Figs. 3, 4). We have pointed to the contrast in elevation at the mountaintop site to the surrounding terrain as a key factor in explaining differences observed at the three sites (Figs. 7, 9, 11). Hopefully these results would stimulate other researchers around the world to also examine the same approaches and factors mentioned in this paper, but the dedicated modeling efforts necessary to do so would be outside the purview of this paper.

*Overall this paper also does not fully address the question of "constraining carbon fluxes", but rather how well different model setups can reproduce the atmospheric concentrations of CO2.*

*To be able to really judge if the models (even the best, 1.3km resolution) are able to e.g. distinguish different prior carbon flux estimates the authors would need to perform a sensitivity study using multiple carbon flux data sets and demonstrate a significant impact at the three sites. Potentially one can consider this study rather a step towards a better use of such data rather than already addressing the question of regional carbon fluxes.*

We agree that the paper is the first step towards addressing the question of regional carbon fluxes. However, we believe that it is a critical first step that needs to be taken in order to use mountaintop $CO_2$ data to constrain regional carbon fluxes.

*After the above and below comments are addressed I would definitely recommend this study for publication in ACP as it will help to better understand limitations of such observations and were future model developments should/could be focussed to eventually be able to constrain carbon fluxes in such regions.*

*Specific comments:*
*Line 44ff: The claim that nearly 70% of the earth land surface is covered by hills or mountains needs to be better validated. This surely depends on the definition for hill or mountain, which is not given here and the cited publication is hard to access (due to the journal it was published in) and the journal has an impact factor below 1.*

We thank the Reviewer for pointing this out. The claim of ~70% of the Earth's land surface as being covered by hills or mountains was attributed to Rotach et al. [2008], but we traced this claim to a book written in the German language. Furthermore, there appears to be no clear definition for what is meant by "hills". Therefore, we decided to revise the statement to just referring to "mountains", which cover about one quarter of the Earth's mountains, citing a readily-accessible UNEP report for this purpose (Blyth et al., 2002).

Blyth, S., B. Groombridge, I. Lysenko, L. Miles, and A. Newton, Mountain Watch: Environmental Change and Sustainable Development in Mountains, UNEP World Conservation Monitoring Centre, 2002.

*The authors also mention that carbon fluxes in complex terrain need to be better understood to quantify carbon flux. It seems you are implying that all mountainous or hilly areas are (too) hard to model?*

We were not necessarily implying all mountainous or hilly areas are difficult to model. We were suggesting that because mountainous areas cover a large fraction of the Earth's land surface and significant amounts of biomass can be found in mountains (e.g., Fig. 1), a better understanding of this under-sampled region is necessary.

*Line 193: The authors refer to a previous publication – nonetheless the key parameters e.g. vertical mixing scheme used should be explicitly given in this publication (e.g. by adding a table in this section).*

We have revised the paper to include key parameters regarding the WRF configuration. The revised sentence reads:
"Comprehensive testing of different WRF settings have been carried out as part of a previous publication (Mallia et al., 2015), and these settings were adopted here: i.e., the MYJ, Grell-

Devenyi Ensemble, and Purdue Lin schemes for parameterizing the planetary boundary layer (PBL), cumulus convection, and microphysics, respectively."

*Line 216: Please specify if the system allows for two-way nesting or not*
Following our testing in Mallia et al. (2015), we have implemented two-way nesting within WRF. This is clarified in the revised text in Sect. 2.2:
"For this study, we ran WRF in a two-way nested mode centered between Utah and Colorado where the RACCOON sites are located (Fig. 2)."

*Line 277: A "fix" is mentioned, but not explained at all. Please consider giving a brief description here rather than referring to the supplement. It seems the daily cycle has just been shifted or were there any more complicated adjustments performed?*
Yes—only the diurnal pattern has been shifted while preserving the 24-hour integrated carbon flux. We have added more information for the reader in the revised text:
"For this paper, we implemented a fix that removed this artifact by detecting these reversed diurnal patterns, adjusting them while preserving the 24-hour integrated carbon flux. See the Supplement and Fig. S3 for details."

*Line 304: Please consider referring to table 1 here so the reader can easily find the height CT data was extracted from.*
We have added a reference to Table 1.

*Line 503ff: The first question is repeated here "How can mountaintop $CO_2$ observations be used to constrain regional scale carbon fluxes, : : :." But the 5 subsections following rather discuss IF such data can be used or how they can be better used. It remains unclear if there is a definitive answer on how to use them.*
Due to the fact that the errors incurred depends on the model resolution, the relationship of the mountaintop site relative to surrounding terrain and emissions, and the quality of the prior fluxes, the definitive answer depends upon each specific situation. Thus we were hesitant to suggest an answer that overgeneralizes. Also see above for the response regarding statements for other mountainous areas.

*Line 530ff: Choosing the appropriate model layer to extract $CO_2$ does indeed introduce a significant additional degree of freedom. The authors suggest other parameters to avoid creating a fudge factor but do not give specific advice here on which tracers could be useful ($^{222}Rn$?). Meteorological data is mentioned but looking at table 2 it seems not at all clear that this would be good parameter or what a cut-off would be. Could you suggest how a suitable proxy could be found?*
These tracers and meteorological data were mentioned later on in the Discussion section:
"We recommend additional tracers to be measured in conjunction with the mountaintop $CO_2$ sites. For instance, combustion tracers such as $C^{14}$ and CO (Levin and Karstens, 2007) have been measured alongside $CO_2$ at mountaintop sites in Europe. Another promising tracer is $Rn^{222}$ (Griffiths et al., 2014), which provides a measure of surface exchange and would help provide constraints on the exchange of air measured at the mountaintop with the surface. Co-located meteorological observations—whether in-situ or remotely-sensed (e.g., radar, sodar, lidar)—to probe atmospheric flows and turbulent mixing would also be of significant value in helping to interpret the tracer observations (Rotach et al., 2014;Banta et al., 2013)."

*Line 581ff: Here the authors report on the practice of not using mountaintop data but it is unclear how this is linked to this specific study as no suggestion is made how to e.g. better use Schauinsland data. This section should be considered for the introduction to motivate why Approach 1, 3, 4, 5 need to be improved.*
We thank the Reviewer for this valuable suggestion. We have followed the Reviewer's suggestion and moved Approach 2 ("Reject mountaintop data") to the Introduction to provide further motivation for the paper.

*Line 595ff: When setting up an inversion system it is common (good) practice to assign proper model errors. This seems not specific to this study and the authors fail to give an estimate of the model error for the three sites discussed here. Please consider removing this section or giving quantitative results for the sites and models investigated here. Of course, the model data mismatch calculated here also depends on flux errors, but the authors can surely use this study to give an upper limit of this combined error (and the difference for different model resolutions).*
We agree with the Reviewer that quantitative results for the sites and models investigated here would be useful. An estimate of these errors is the RMSE (root-mean-square-error) calculated for each site and model setup. We now direct the reader to the RMSE values shown in the top right-hand corner of the time series plots found at the beginning of this Response.

We thank Reviewer #3 for the constructive criticism of the Discussion paper. The reviewer's comments are shown below in *italics*, while our point-by-point responses are indicated as un-italicized.

**Reviewer #3**

*The major concern is that how different the $CO_2$ simulations and observations are compared from day to day, particularly when finer scale models are used as suggested? Averaged results might miss some crucial information hidden in the discrepancies between model simulations and site observations in present study.*

We agree with the Reviewer that beyond the diurnal timescale, there is day-to-day variability in the model behavior that lead to variations in $CO_2$ model errors. We have now included in the revised paper the time series of $CO_2$ model errors at SPL, NWR and HDP (see below) with correlations of the $CO_2$ errors with various meteorological variables (geopotential height and its E⇔W gradient, U- and V-winds, and the windspeed). The time series plots and a Table of the correlations of the $CO_2$ errors with these meteorological variables will be added to the Supplement.

Errors at multi-day timescales are the most strongly correlated with different meteorological variables, depending on which is being examined: V-wind, U-wind, and geopotential height gradient for HDP, SPL, and NWR, respectively. The fact that errors are correlated with different meteorological variables depending on the site location points to a complexity that can only be unraveled with a substantial expansion of the paper. This could potentially be a subject for a future paper. This complexity is in contrast to the average diurnal biases that can in large part be linked to the underlying resolution of modeled terrain, which the current paper focuses on.

[Figure]

[Figure]

*The related studies focusing on mountaintop such as this one are important and rarely explored. I would recommend this manuscript for publication after above concern is addressed.*

*Specific comments:*

*Line 146: How is the vegetation covered at these mountain stations? Are these monitoring sites far above the tree line? Local influences on $CO_2$ due to the surrounding plant cover at these sites may need to be considered.*

Both HDP and NWR are above the tree line; SPL has a few sparse trees around the site. The RACCOON mountaintop observations have been filtered to remove local influences and to extract values that are more regionally representative, following the work in Brooks et al. (2012). These filtered observations were the ones used in the paper. We have added this reference and modified the text to clarify this point:

"We applied filtering to the mountaintop $CO_2$ observations to remove local influences and to extract values that are more regionally representative (Brooks et al., 2012). Observations were filtered out in which the within-hour standard deviation is greater than 1.0 ppm or when the differences between the top two inlets are greater than 0.5 ppm, which indicate periods when significant influences that are highly localized to the site are affecting the observations."

Brooks, B.-G. J., Desai, A. R., Stephens, B. B., Bowling, D. R., Burns, S. P., Watt, A. S., Heck, S. L. and Sweeney, C.: Assessing filtering of mountaintop $CO_2$ mole fractions for application to inverse models of biosphere-atmosphere carbon exchange, Atmos. Chem. Phys., 12, 2099–2115, 2012.

*Lines 296-297: How much percent is contributed to the wildfires during the study period?*
The exact percentage depends on the quantify under consideration: gross versus net fluxes, and the percentage would also depend upon the time of the day. Regardless, the percent is very small and is visually apparent from Fig. S4. We added an additional reference to Fig. S4 to clarify this point:
"Contributions from anthropogenic and wildfire emissions, on average, to the mean $CO_2$ diurnal cycle observed at all the mountain sites were secondary in comparison to the biosphere (Fig. S4). In particular, the wildfire contributions were episodic and averaged out to negligible contributions over Jun~Aug 2012 (Fig. S4)."

*Lines 301-302 and Fig. 3: Fig. 3 illustrates the 3-month averaged diurnal cycles of the results from individual models and observations. Do they show similar diurnal cycles every day during these months? Consider addressing the variabilities of the diurnal cycles (e.g. error bars) in the supplement.*
To illustrate the variability in the diurnal cycles we have added error bars to the average diurnal cycles shown in Fig. 3.

*Fig. 4: Has the GDAS_1-deg_ASL in Fig. 4 been adjusted with the biospheric fluxes? If not, I would recommend drawing it as gray line and dot as presented in Fig. 3.*
Fig. 4 is showing the average diurnal cycle of the footprint totals, which result solely from the simulated atmospheric transport. Thus biospheric fluxes are not incorporated into Fig. 4 and therefore the dashed + dot scheme paralleling Fig. 3 is not necessary here.

*Fig. S5: Two CT Level 1 (dark blue) and two CT Level 2 (blue)?*
We are not sure about the exact meaning of the Reviewer's comment here. The CT levels are color-coded as a gradation from dark blue (Level 1) to dark green (Level 8); Level 2 is indeed colored as blue.

*Fig. S5 and Table 1: The differences between the site and model altitude (Table 1) seem to be generally associated with the $CO_2$ biases. For instance, larger differences such as WRF-12km, GDAS, and Carbon Tracker showed larger discrepancies in diurnal cycles of $CO_2$ at HDP. What*

*if the model altitude fixed as same as the mountain site? Could it be better correlated to the observational $CO_2$ data?*

We are in full agreement with the Reviewer that $CO_2$ biases are associated with differences in site versus model altitudes, stemming from increasing discrepancies with terrain as the spatial resolution is degraded. This point is discussed extensively in Sect. 3.2.2. Adjusting the vertical level within the model is an approach that we explored Sect. 4.1 ("Approach 1: Adjust vertical level of simulations from which to compare against observed values")

*Lines 336-380: Are these trajectories in a good agreement for each day during the study period both in nighttime and afternoon hours?*

We have generated the time series of $CO_2$ error at different days during the study and examined the correlation between the errors with other meteorological variables to address this question. See details at the top of this response for more details.

[revised manuscript text omitted]

[Mton of carbon on 0.5°x0.5° grid]

**Fig. 1**

**WRF Domains**

[Figure]

**Fig. 2**

[Figure]

**Fig. 3**

[Figure]

**Fig. 4**

[Figure]

**0900 UTC (0200 MST)**

HDP ave footprint: WRF-1.3km (AGL)

HDP ave footprint: WRF-4km (AGL)

HDP ave footprint: WRF-12km (AGL)

HDP ave footprint: GDAS-1° (ASL)

Nighttime (1900~0700 MST)

Daytime (0700~1900 MST)

Trajectory in PBL

**Fig. 5**

[Figure]

**Fig. 6**

**HDP:  Mean 3D Trajectory of Stochastic Particles & PBL ht for Different Runs**
**0900 UTC (0200 MST)**

[Figure]

**Fig. 7**

**HDP:  Mean 3D Trajectory of Stochastic Particles & PBL ht (above ground level)**
**0900 UTC (0200 MST)**

[Figure]

**Fig. 8**

**SPL:  Mean 3D Trajectory of Stochastic Particles & PBL ht for Different Runs**
**0900 UTC (0200 MST)**

[Figure]

**Fig. 9**

**SPL:  Mean 3D Trajectory of Stochastic Particles & PBL ht (above ground level)**
**0900 UTC (0200 MST)**

[Figure]

**Fig. 10**

**NWR: Mean 3D Trajectory of Stochastic Particles & PBL ht for Different Runs**
**0900 UTC (0200 MST)**

[Figure]

**Fig. 11**

**NWR:  Mean 3D Trajectory of Stochastic Particles & PBL ht (above ground level)**
**0900 UTC (0200 MST)**

[Figure]

**Fig. 12**

**Mean 3D Trajectory of Stochastic Particles & PBL ht**
**GDAS-1° (ASL)**

[Figure]

**Fig. 13**

**SUPPLEMENTAL INFORMATION:**

**How can mountaintop CO$_2$ observations be used to constrain regional**
**carbon fluxes?**

John C. Lin[1], Derek V. Mallia[1], Dien Wu[1], Britton B. Stephens[2]

[1]Department of Atmospheric Sciences, University of Utah, Salt Lake City, Utah 84112, USA
[2]Earth Observing Laboratory, National Center for Atmospheric Research, Boulder, Colorado
80301, USA
*Correspondence to*:  John C. Lin (John.Lin@utah.edu)

Manuscript Submitted to Atmospheric Chemistry and Physics

**Fig. S1**
Three-dimensional plots of the terrain over a domain of ~1$^o$×1$^o$ surrounding the NWR
site, as resolved by the WRF 1.3-km model.  The NWR site is indicated by the triangle.
A small subsample of the numerous stochastic trajectories simulated by STILT, driven by
WRF started at 2100 UTC (1400 MST), are drawn as black lines.  Also shown is the
average back trajectory (pink), derived by averaging locations of the stochastic
trajectories.  In addition, the PBL heights averaged along the backtrajectory are shown as
the blue line.

**Adjusting the CT-2013b diurnal cycle**

In the CarbonTracker assimilation process, attempts to match $CO_2$ observations could result in "dipoles" in scaling factors between nearby ecoregions, leading to negative fluxes even at night (Fig. S2a). While respiration can occur during the day when vegetation is under stress (e.g., droughts), photosynthetic uptake (negative fluxes) at night, in the absence of sunlight, is biologically unphysical. In order to correct the reversed diurnal cycle seen in CarbonTracker, a reversal had to be first detected within CarbonTracker for the selected grid cell for a given day. Once the reversal was detected, the sign of the biospheric flux was flipped. The positive flux was then adjusted so that the net flux for the selected gridcell for the given day was equal to 0. Finally, the negative flux was adjusted so that the final net flux was equal to the original net flux, which preserved the total net flux for the day (Fig. S3). The resulting biosperic flux pattern can seen in Fig. S2b.

**Fig. S2**

Mean biospheric fluxes from Jun~Aug 2012 averaged between 0600~0900 UTC (2300~0200 MST). (a) Biospheric fluxes for the unmodified CarbonTracker flux fields and (b) biospheric fluxes for the adjusted CarbonTracker flux fields. The black circle represents HDP, the black diamond represents SPL, and the black star represents NWR.

**Fig. S3**

Schematic showing the adjustment of erroneous diurnal pattern in biospheric flux within CarbonTracker (red line), with nighttime uptake, to a corrected biospheric flux (green line). The dashed line represents a flux of 0.

**Fig. S4**
Average contributions to $CO_2$ variations at HDP, SPL, and NWR from biospheric, anthropogenic, and wildfire fluxes at different times of the day between Jun~Aug 2012 as simulated by STILT, driven with WRF-1.3km winds.  Also shown are the observed variations, calculated by subtracting out the STILT-derived background (see Sect. 2.3).

**Fig. S5**
Time series of $CO_2$ errors (Simulated – Observed) at the HDP, SPL, and NWR sites during the study period (Jun~Aug 2012) for the different model configurations—i.e., WRF-1.3km AGL, 4-km AGL, 12-km AGL, and GDAS-ASL.  The thin lines denote $CO_2$ errors calculated at high frequency, at 3-hourly time spacing.  The thick lines represent the $CO_2$ errors smoothed with a 4-day centered running average that will be correlated with other meteorological variables (Table S1). The bias and root-mean-square error (RMSE) reported on the top-right hand box are calculated based on the 3-hourly time series.  The gap in the earlier part of July at NWR is due to missing observations.

**Fig. S6**
Mean $CO_2$ concentrations extracted from the bottom 8 levels of CarbonTracker, in the respective gridcells where the HDP, SPL, and NWR sites are located.  The mean model heights of the bottom 8 levels are (in meters AGL):  25, 103, 247, 480, 814, 1259, 1822, 2508. The concentrations interpolated to the heights of the 3 sites are indicated by the orange dashed line.  The observed values are drawn in black, with unfiltered data (dashed) and after applying the filter for removing local influences (solid;  Sect. 2.1).

**Fig. S7**
The average footprint (shown in $\log_{10}$) for the SPL at 0200 MST (0900 UTC), gridded at $0.1^{o} \times 0.1^{o}$.  The site is denoted as a triangle.  The average back trajectory (averaged over the stochastic STILT trajectories) is drawn as a line, with points indicating trajectory locations every hour, as the trajectory moves back from the site indicated as points.  Magenta parts of the trajectory refer to the nighttime (1900~0700 MST), while pink portions indicate the daytime (0700~1900 MST).  Parts of the trajectory are shaded with blue when it is found below the average height of the PBL along the trajectory.

**Fig. S8**
Similar to Fig. S7, but for 1400 MST (2100 UTC).

**Fig. S9**
Similar to Fig. S7, but for the NWR site.

**Fig. S10**
Similar to Fig. S9, but for 1400 MST (2100 UTC).

**Fig. S11**
Three dimensional plots of the terrain over a domain of ~1$^{o}$×1$^{o}$ surrounding HDP, as resolved by the WRF and GDAS models at various grid spacings. Also shown is the average back trajectory, derived by averaging locations of the numerous stochastic trajectories simulated by STILT, driven by the various WRF meteorological fields and the global GDAS field. Back trajectories were started from HDP at 1400 MST (2100 UTC). Points indicate trajectory locations every hour, as the trajectory moves back from the site indicated as points. Magenta portions of the trajectory refer to the nighttime (1900~0700 MST), while pink portions indicate the daytime (0700~1900 MST). In addition, the PBL heights averaged along the backtrajectory are shown as the blue line.

**Fig. S12**
Similar to Fig. S11, but for SPL.

**Fig. S13**
Similar to Fig. S11, but for NWR.

**Table S1**

Correlation coefficients between $CO_2$ errors smoothed with a 4-day centered running average (Fig. S5) and potential explanatory meteorological variables observed near the HDP, SPL, and NWR sites. The smoothing window of 4-days was selected to focus on synoptic scale meteorological changes. The meteorological observations come from radiosondes launched at 00-UTC and 12-UTC from the following airports: Salt Lake City, Grand Junction, and Denver for the HDP, SPL, and NWR sites, respectively. The meteorological variables are extracted at the 500-hPa level and include the observed geopotential height (GPH), geopotential height gradient between NWR and HDP (NWR – HDP), observed windspeed, as well as the U- and V- components of the observed wind vector. The GPH time series is processed by subtracting its 20-day running average (centered) to remove trends and seasonal variations and then smoothed with a 4-day running average. Pearson correlation coefficients are reported here; coefficients with lower statistical significance ($p>0.05$) are not shown and indicated with "-" in the Table.

| | HDP | | | | SPL | | | | NWR | | | |
|---|---|---|---|---|---|---|---|---|---|---|---|---|
| | *WRF 1.3-km* | *WRF 4-km* | *WRF 12-km* | *GDAS* | *WRF 1.3-km* | *WRF 4-km* | *WRF 12-km* | *GDAS* | *WRF 1.3-km* | *WRF 4-km* | *WRF 12-km* | *GDAS* |
| **GPH** | 0.35 | 0.57 | - | 0.39 | - | - | 0.17 | - | - | - | - | -0.19 |
| **GPH gradient** | -0.37 | -0.39 | - | - | 0.20 | 0.25 | 0.57 | - | -0.71 | -0.69 | -0.64 | -0.66 |
| **U wind** | - | - | -0.30 | - | 0.60 | 0.64 | 0.54 | 0.49 | 0.30 | 0.33 | 0.20 | 0.37 |
| **V wind** | -0.53 | -0.49 | - | - | - | 0.19 | 0.49 | 0.24 | -0.52 | -0.52 | -0.44 | -0.49 |
| **Windspeed** | -0.16 | -0.24 | -0.18 | - | 0.43 | 0.49 | 0.59 | 0.34 | - | - | - | - |

[Figure]

**2100 UTC (1400 MST)**

**Fig. S1**

[Figure]

**Fig. S2**

[Figure]

**Fig. S3**

**Average Diurnal Contributions from Different CO₂ Sources**

[Figure]

**Fig. S4**

[Figure]

**Fig. S5**

[Figure]

**Fig. S6**

**SPL ave footprint: WRF-1.3km (AGL)**

[Figure]

log10(footprint)

**SPL ave footprint: WRF-4km (AGL)**

[Figure]

log10(footprint)

**0900 UTC**
**(0200 MST)**

**Nighttime
(1900~0700 MST)**

**Daytime
(0700~1900 MST)**

**Trajectory in
PBL**

**SPL ave footprint: WRF-12km (AGL)**

[Figure]

log10(footprint)

**SPL ave footprint: GDAS-1° (ASL)**

[Figure]

log10(footprint)

**Fig. S7**

**SPL ave footprint:  WRF-1.3km (AGL)**

[Figure]

**SPL ave footprint:  WRF-4km (AGL)**

[Figure]

**2100 UTC
(1400 MST)**

**Nighttime
(1900~0700 MST)**

**Daytime
(0700~1900 MST)**

**Trajectory in
PBL**

**SPL ave footprint:  WRF-12km (AGL)**

[Figure]

**SPL ave footprint:  GDAS-1° (ASL)**

[Figure]

**Fig. S8**

**NWR ave footprint:  WRF-1.3km (AGL)**

[Figure]

log10(footprint)

**0900 UTC
(0200 MST)**

**NWR ave footprint:  WRF-4km (AGL)**

[Figure]

log10(footprint)

**Nighttime
(1900~0700 MST)**

**Daytime
(0700~1900 MST)**

**Trajectory in
PBL**

**NWR ave footprint:  WRF-12km (AGL)**

[Figure]

log10(footprint)

**NWR ave footprint:  GDAS-1º (ASL)**

[Figure]

log10(footprint)

**Fig. S9**

**NWR ave footprint:  WRF-1.3km (AGL)**

[Figure]

**2100 UTC
(1400 MST)**

**NWR ave footprint:  WRF-4km (AGL)**

[Figure]

**Nighttime
(1900~0700 MST)**

**Daytime
(0700~1900 MST)**

**Trajectory in
PBL**

**NWR ave footprint:  WRF-12km (AGL)**

[Figure]

**NWR ave footprint:  GDAS-1° (ASL)**

[Figure]

**Fig. S10**

[Figure]

**HDP:  Mean 3D Trajectory of Stochastic Particles & PBL ht for Different Runs**
**2100 UTC (1400 MST)**

**Fig. S11**

[Figure]

**SPL: Mean 3D Trajectory of Stochastic Particles & PBL ht for Different Runs**
**2100 UTC (1400 MST)**

WRF-1.3km (AGL)

WRF-4km (AGL)

WRF-12km (AGL)

GDAS-1° (ASL)

**Fig. S12**

**NWR: Mean 3D Trajectory of Stochastic Particles & PBL ht for Different Runs**
**2100 UTC (1400 MST)**

[Figure]

**Fig. S13**